# A nanofluidic knot factory based on compression of single DNA in nanochannels

Susan Amin[1], Ahmed Khorshid[1], Lili Zeng[1], Philip Zimny[2] & Walter Reisner[1]

Knots form when polymers self-entangle, a process enhanced by compaction with important implications in biological and artificial systems involving chain confinement. In particular, new experimental tools are needed to assess the impact of multiple variables influencing knotting probability. Here, we introduce a nanofluidic knot factory for efficient knot formation and detection. Knots are produced during hydrodynamic compression of single DNA molecules against barriers in a nanochannel; subsequent extension of the chain enables direct assessment of the number of independently evolving knots. Knotting probability increases with chain compression as well as with waiting time in the compressed state. Using a free energy derived from scaling arguments, we develop a knot-formation model that can quantify the effect of interactions and the breakdown of Poisson statistics at high compression. Our model suggests that highly compressed knotted states are stabilized by a decreased free energy as knotted contour contributes a lower self-exclusion derived free energy.

[1] Department of Physics, McGill University, 3600 rue université, Montréal, QC H3A 2T8, Canada. [2] Department of Biomedical Engineering, McGill University, 3775 rue université, Montréal, QC H3A 2B4, Canada. Correspondence and requests for materials should be addressed to W.R. (email: reisner@physics.mcgill.ca)

Knots naturally exist in DNA, proteins, umbilical cords, and catheters[1,2]. Knots can form when an initially linear chain passes its loose free ends through one or multiple loops on the same chain, giving rise to a knot if the polymer is subsequently cyclized. For example, random cyclization of linear DNA in bulk[3] forms knotted chains with low probability; or knots can be directly tied via optical tweezers[4]. Chain compaction, induced via either spatial confinement, compression, or molecular crowding[5], tends to enhance the tendency for chains to self-entangle, and thus enhances knotting probability. Knot formation on DNA is a particular challenge in biology, due to high degree of compaction experienced by packaged genomes, and is consequently tightly regulated by enzymes like topoisomerases and recombinases that remove knots by breaking and rejoining of either single or double strands[2]. An extreme example is the high level of compaction experienced by viral genomes[6], resulting in a correspondingly high knotting probability for DNA extracted following capsid rupture[6]. Knots on genomic DNA in nanofluidic systems interfere with mapping by preventing complete linearization of contour stored in the knot, giving rise to an artifact resembling a deletion[7].

Consequently, there has been intense theoretical focus on knot production mechanisms[2] and physics of confined knots[8–11]. Yet, while single-molecule techniques for knot sensing are advancing rapidly[12], and single-knot diffusion and size dynamics have been explored[4,13], systematic experimental studies probing conditions enhancing knot formation in microscopic systems are limited. Knotting in DNA extracted from the P4 phage system has been extensively studied, but an in vivo system has inherent disadvantages, including a fixed parameter space and difficulty of determining whether knotting occurs inside the capsid or following rupture. There have been reports of knot formation in nanochannels;[14] coil collapse in an AC field has been used to induce knotted and self-entangled states of a single chain[15], but these experiments did not systematically quantify knot formation as a function of compaction. Understanding of knot-formation in microscopic chains is framed[2,15] by a classic experiment involving tumbling of a macroscopic string inside a rotating box[16]. In this experiment, knots were formed when successive tumbles drove parallel concentrically coiled strands near the chain ends to cross. At low agitation times, knot formation was observed to be kinetically limited; at longer agitation times, the knotting probability saturated at a value that approached unity for longer, highly flexible strings[2]. An intriguing question is whether experiments probing knot formation in microscopic chains might reveal a similar kinetically limited regime at low times and a saturating knotting probability at long-times.

Here, we introduce a knot factory on chip using low Reynolds number flow to compress single DNA molecules against slit-barriers in nanochannels (Fig. 1a–f). The chain is initially extended (Fig. 1d). After compression (Fig. 1e), the flow is released and the DNA molecules relax (Fig. 1f); knots are present along the relaxed DNA, visualized as sharply localized regions of high intensity on the extended molecule (Fig. 1g–j). The knot-factory enables efficient knot formation and detection in an in vitro system where all parameters are well controlled, guiding

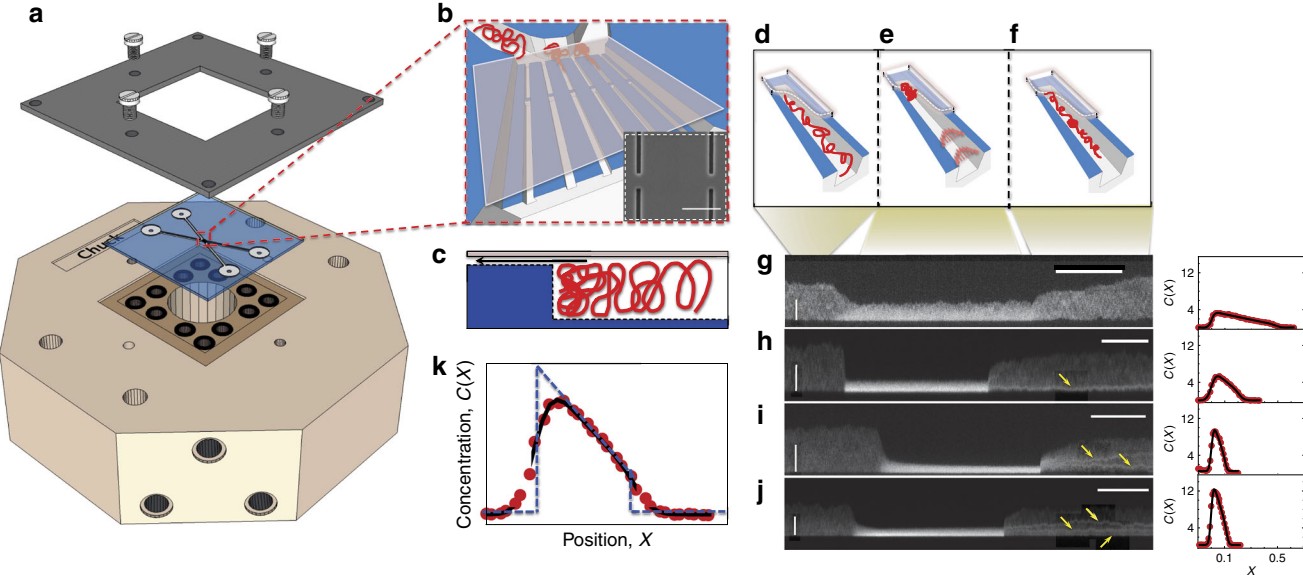

**Fig. 1** Device concept and experimental set-up. **a** The nanofluidic device is mounted on a chuck containing inputs for application of pneumatic pressure to transport DNA molecules in micro and nanochannels and enable hydrodynamic compression against slit-barriers. **b** A magnified view of the chip center. The device is composed of two 1-μm deep loading channels spanned by a nanochannel array with blunt-ended barriers in the channel centers. A 30-nm deep slit, etched over the array, allows for solvent to escape while preventing the passage of large DNA molecules. Inset: an SEM image of the nanochannels with barrier (the scale bar is 3-μm). **c** A magnified cross-sectional view of a nanochannel at the device center showing the slit-barrier. The black arrow depicts the flow direction through the slit. **d–f** A 3D cartoon showing the process of knot-formation detailing **d** DNA confinement, **e** compression against the barrier via hydrodynamic flow induced by applying a pressure-drop across the nanochannels and **f** relaxation of a knotted chain. The red arrows in **e** depict the flow velocity profile during compression. **g–j** Examples of kymographs for knot-formation events with increasing compression degree. Intensity along the nanochannel (vertical axis, scale bar10 μm) is plotted vs. time (horizontal axis, scale bar 10 s). Each molecule is compressed, held at a minimum extension for a waiting time $t_w$, and then relaxed. Normalized chain concentration profiles corresponding to the kymographs are illustrated on the right. **g** No knot is formed; **h** one knot is formed; **i** two knots and **j** three knots are formed. The yellow arrows depict the knot locations. The second bright spot in **h** does not maintain its size and unravels in the chain mid-section shortly after pressure release so we do not count it as a knot[20]. **k** Normalized chain concentration profile $C(X)$, averaged over waiting time at compressed state; red circles are experiment; black line is a fit to a linear ramp concentration profile (Eq. 1) convolved with a Gaussian point-spread function (Supplementary Note 3D). The blue-dashed line shows an estimate of the real, i.e., prior to convolution, concentration profile estimated from the fit

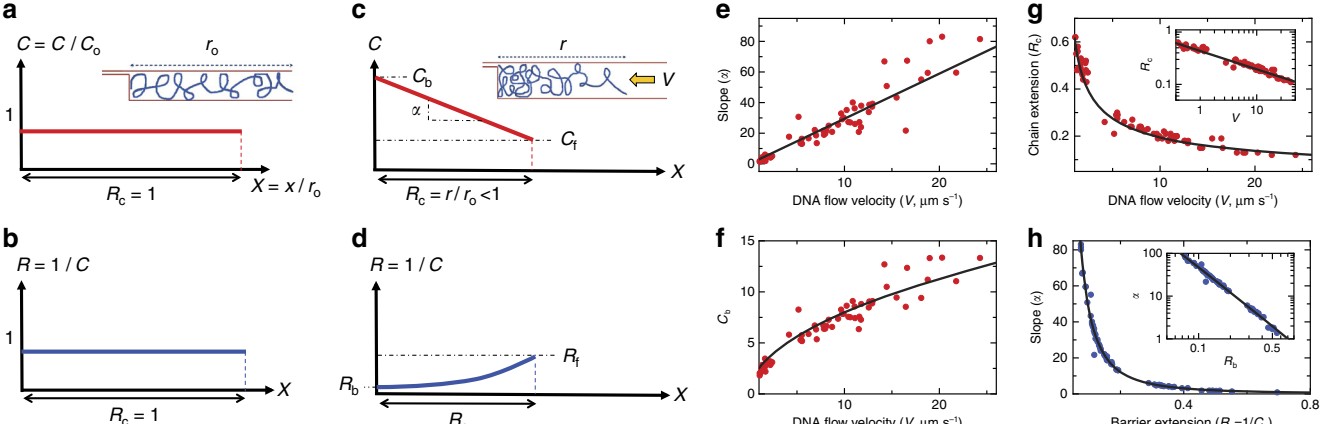

**Fig. 2** Relation between the parameters defined in concentration profile. **a** The nanochannel confined chain in no-flow equilibrium has a uniform concentration profile $C(X) = 1$. Inset: schematic of no-flow equilibrium chain with extension $r_o$. **b** Local extension $R(X) \equiv 1/C(X)$ for chain in no-flow equilibrium. **c** When a flow $V$ is applied, in the long-time limit the chain reaches a steady-state with a concentration profile that ramps linearly towards the barrier: $C(X) = C_b - \alpha X$. Inset: schematic of chain in flow-constrained equilibrium with extension $r$. **d** Local extension $R(X) \equiv 1/C(X)$ for chain in flow-constrained equilibrium. Note that for a uniform profile $R_b = R_c$. **e** Profile slope $\alpha$, **f** barrier concentration $C_b$, and **g** chain extension $R_c$ vs. $V$ with fits to scaling relations predicted by piston theory (**e** $\alpha \sim V$, **f** $C_b \sim \sqrt{V}$ and **g** $R_c \sim 1/\sqrt{V}$). **h** Combining data in **e** and **f** yields $\alpha$ vs. $R_b$, described well by the scaling $\alpha \sim 1/R_b^2$. The insets in **g** and **h** give the results on a log–log-scale

development of models to quantify conditions favoring knot production. In particular, we can measure knot-formation probability as a function of compression and probe knot formation kinetics by relaxing the chain after a well-defined waiting period in the compressed state. By measuring knot position along the chain a short time after pressure release, we can gain insight into the knot spatial distributions. Finally, we can access conditions where more than one knot is formed, enabling investigation of the formation of composite knot states. Our results suggest that strong interaction between prime knots exist in composite knot states, leading to a breakdown of independent Poisson knot-formation statistics observed for extended chains at equilibrium in the absence of compression[17]. To rationalize our findings, we argue that the compressed chain is in a steady state with zero segmental current, equivalent to a state of inhomogeneous equilibrium, so that a generalized free energy can be developed to quantify the probability of knot-formation. This approach complements existing theories for knot-production by clarifying knotting free energy landscapes for compressed chains and explains our observations if topological barriers for forming knots are sufficiently small relative to thermal or flow-driven agitation for the chain to sample knotted states over measurement time-scales.

## Results

**Assessment of knotted states.** The knotting state can be assessed by counting the number of knots present. Figure 1g–j gives example kymographs for individual compression–relaxation events at different degrees of compression. For very high compression we observe that more than one knot can be formed (Fig. 1i, j). Knots can be distinguished from other topological events such as folds[18] or trivial knots (including entangled segments and complex unknots, like slip-knots[19], that do not possess true knot topology) as knots formed on semiflexible chains quickly adopt a characteristic compact and time invariant structure and are only removed when they diffuse to the chain ends[8,20–22]. In contrast, trivial knots or unknots, such as folds or entangled regions, are expected to gradually unravel in mid-chain or at the chain edges under the influence of entropic forces driving contour to less confined regions[18]. Thus, in contrast to other topological events, knots are objects that once formed on

the polymer: (1) are persistent, localized and bright features; (2) do not exhibit large-scale size fluctuations after reaching their final state; and (3) can unravel only at the molecule ends.

**DNA concentration profile during compression.** The nanoscale dimensions of our channels give rise to ultra-low Reynolds number hydrodynamics (Re $\sim 10^{-8}$) that necessitate rigorously laminar and steady streamlines in the presence of a constant pressure drop[23]. Note that formation of nano-vortices at the slit-barrier requires a Reynolds number exceeding Re = 0.055[24]. In contrast to[15], we do not apply an external electric field. We estimate that any streaming potential difference[23] across the nanochannel resulting from our flow is <1 mV, affecting the DNA velocity by <1% (see Supplementary Note 3A).

We find the laminar flow leads to physics analogous to that of our optical piston experiments[25,26], where an optically trapped bead is used as a sliding gasket to compress single double-stranded DNA molecule with fixed velocity $V$ (see Supplementary Note 3B for extended discussion). Like the sliding-gasket experiments, during the first phase of compression, a "shock-wave" of segmental concentration builds up at the molecule edge abutting the barrier[26] (Fig. 1g–j). In this phase, the position of the molecule edge opposite the barrier (the free edge) is unconstrained and has constant speed $V$, a measure of the buffer flow speed in the channel (see Supplementary Figure 2 and Supplementary Note 3C for measurement of $V$). The second phase begins when the shock-wave reaches the free edge. In this second phase, the laminar flow forces the chain immobile against the slit barrier with forces due to the osmotic pressure gradient everywhere balancing hydrodynamic forces so that the net polymer current is zero (i.e., zero net movement of Kuhn segments). This zero current steady-state is equivalent to a state of inhomogeneous or force-constrained equilibrium[27]. In this state, the compressed molecule, spanning the range from $x = 0$ to $x = r$, adopts a ramped concentration profile (Figs. 1k and 2c). Sliding gasket theory suggests the ramp is linear; in terms of the normalized variables $C \equiv c(x)/c_o$, $X \equiv x/r_o$ and $R_c = r/r_o$ the ramp has the form[26]

$$C(X) = C_b - \alpha X, \qquad (1)$$

with $X$ ranging from zero to $R_c$. $r_o$ and $c_o$ are respectively the molecule equilibrium extension and concentration in the absence of flow, determined via a standard fitting model based on the convolution of a box with a Gaussian point-spread function[28] (see Supplementary Note 3D, note $r_o = 14.3 \pm 0.3 \, \mu m$ for our channels). $r$ represents the molecule extension at compressed state, held for a waiting time $t_w$ (Fig. 1e, see Supplementary Note 3E for the exact definition of waiting time). The quantity $C_b \equiv C(0)$ is the (maximal) concentration at the slit barrier and $\alpha$ is the ramp slope (Fig. 2c). Equation 1, once convolved with a Gaussian point-spread function (see Supplementary Note 3D), describes experiment well (Fig. 1k). The parameters $\alpha$, $C_b$ and $R_c$ are extracted from the experimental profiles via fitting to the convolved linear ramp (Fig. 1k) and plotted as a function of $V$ (Fig. 2e–g). The plots show that the $V$-dependence is indeed consistent with gasket theory, which predicts the scalings $C_b \sim \sqrt{V}$, $\alpha \sim V$ and $R_c \sim 1/\sqrt{V}$[25,26] (see Supplementary Note 3B for a review of the derivation of these scalings).

In addition, we choose to introduce a local extension $R(X) \equiv 1/C(X)$ (Fig. 2b, d). The local extension measures how locally compressed ($R < 1$) the chain is relative to the no-flow equilibrium where $C = R = 1$ everywhere along the chain. In particular, we use the local extension at the slit barrier, or barrier extension, defined by $R_b \equiv 1/C_b$ (Fig. 2d), to parameterize the compression profile in lieu of $V$ or $R_c$. The barrier extension has useful properties; like $R_c$ it becomes strictly smaller with increasing compression, is proportional to, but less than $R_c$ ($R_b/R_c = 0.62 \pm 0.05$, see Supplementary Note 3B) and directly characterizes chain properties at the slit barrier where knots are found with highest probability. Figure 2h combines the data in Fig. 2e, f and gives $\alpha$ as a function of $R_b$; these data are well described by the gasket scaling $\alpha \sim 1/R_b^2$.

**Knotting probability measurement**. The time-dependent knotting probability can be described by introducing constant transition rates $k_{ij}$ from a state with $i$ knots to a state with $j$ knots (Fig. 3a), resulting in a set of coupled rate equations. Figure 3b gives knotting probability for single and multiple knot states as a function of waiting time. The rate equations are solved (see Supplementary Note 4) for the time-dependent probabilities and fitted to the experimental results. The knotting probability rises with $t_w$ and then asymptotes to a constant value at long-times ($t > 17 \, s$), suggesting a gradual equilibration of the knotting state. This equilibration time-scale compares on order of magnitude to the extensional relaxation time of confined T4 DNA in channels of our size in no-flow equilibrium (~10 s, obtained from scaling values for the $\lambda$-DNA relaxation time in[29] to T4). Comparable transition rates for the no-knot to one-knot transition and the one-knot to two-knot transition suggest that the presence of an existing knot does not alter the energy barriers involved in forming the second knot ($k_{01} = 0.21 \pm 0.02 \, s^{-1}$ and $k_{12} = 0.19 \pm 0.13 \, s^{-1}$).

Figure 4a gives measurements of knotting probability as a function of $R_b$. The equilibrium knotting probability increases as $R_b$ decreases. In particular, the one-knot states increase in frequency until reaching a peak at around $R_b \approx 0.12$. The two-knot formation probability rises and becomes equal to the one-knot formation probability at $R_b \approx 0.09$. We also observe a very small number of three-knot events (two in total).

Micheletti and Orlandini[17] suggest that formation of composite knots in nanochannel-confined DNA should arise from independent knotting events along the chain, leading to a description via Poisson statistics. In particular, for a chain in no-flow equilibrium such as studied by Micheletti and Orlandini, the Poisson model suggests that the probability of forming a

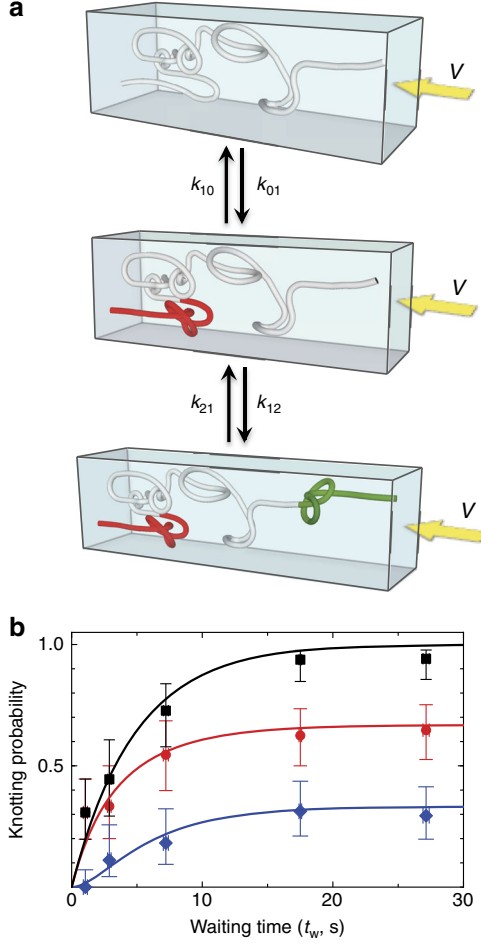

**Fig. 3** Knot formation kinetics. **a** The probability for finding no knots (top), forming 1 knot (middle), and 2 knots (bottom) are related through the transition rates $k_{ij}$ from a state with $i$ knots to a state with $j$ knots. The transition rates define a set of coupled rate equations (see Supplementary Note 4). **b** The probability of knot formation as a function of waiting time at average barrier extension $R_b \approx 0.13$ with fits to the kinetic model. The black squares give experimental measurements for total probability of forming an event with any number of knots. The red circles and blue diamonds give respectively measurements of one-knot and two-knot event probabilities. The continuous lines represent the fits to the time-dependent knotting probabilities predicted by the rate equations. Each data point is determined from the average result of ~10–15 events. The vertical error bars on probability have been calculated using a Wilson-score interval with a one-sigma confidence interval[47] (See Supplementary Note 13); the horizontal error bars show the error on the mean for $t_w$

composite knot based on $m$ number of prime knots of the same topology is governed by

$$P_m = n^m \frac{e^{-n}}{m!} \tag{2}$$

with $n = \frac{L}{L_0}$ where $L$ is the DNA contour length and $L_0$ a characteristic length scale depending on the channel width $D$. While the concentration profile is uniform for a chain in no-flow equilibrium, concentration uniformity is not required for Poisson statistics to hold; Poisson statistics requires only that the prime knots form independently. In an inhomogeneous Poisson process[30], the knot formation probability can vary along the chain, leading to a distribution identical to Eq. 2 but with $n$ expressed as an integral of the varying knot formation probability along the chain. For both the uniform and non-uniform cases, we

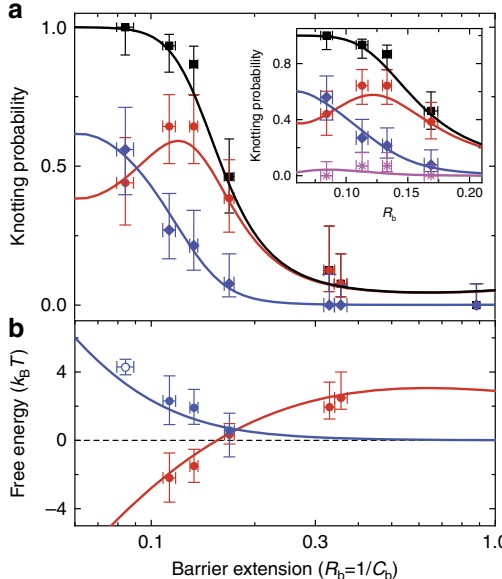

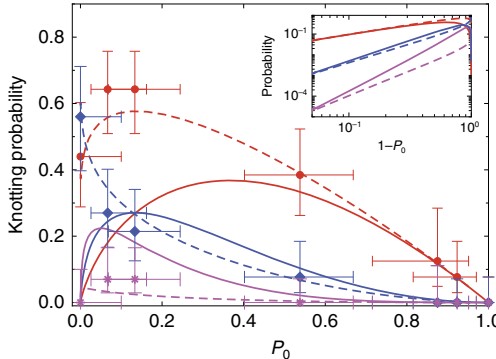

**Fig. 5** Poisson distribution compared with experimental data. The red circles, blue diamonds and magenta stars respectively give measured probabilities for forming one, two and three knots. Each data point is determined from the average result of ~10–15 events. The vertical error bars on probability have been calculated using a Wilson-score interval with a one-sigma confidence interval[47] (See Supplementary Note 13); the horizontal error bars represent the error on the mean for $R_b$ measurements for the corresponding events. The solid lines indicate predictions of pure Poisson statistics (i.e., following from Eq. 3); the dashed lines indicate the predictions of the free energy model, with the red, blue, and magenta curves, respectively corresponding to $m = 1$, $m = 2$ and $m = 3$. The inset, which shows the same theory curves on a log–log scale vs. $1 - P_0$, shows that the free energy model asymptotes to the Poisson description when the compression becomes very low and the profile approaches no-flow equilibrium

**Fig. 4** Knotting probability and free energy assessment. **a** The probability of knot formation as a function of barrier extension (y-scale linear, x-scale log). The inset in **a** illustrates the predicted probability of knotting for a sample space including three-knot formation probability (linear-linear scale). The black squares give experimental measurements for total probability of forming an event with any number of knots. The red circles and blue diamonds give respectively measurements of one-knot and two-knot event probabilities. The magenta stars in the inset give probability measurements for three-knot events. The continuous lines indicate fits to the free energy model. **b** Free energy of single knot states $F_{tot}(1, R_b)$ (red, circles) and two-knot interaction free energy $F_{2,tot}^{int}(R_b)$ (blue, circles) deduced assuming the profile is in a state of inhomogeneous equilibrium, with theoretical overlay using same fitting parameters for **a**. Each data point is determined from the average result of ~10–15 events. The vertical error bars on probability have been calculated using a Wilson-score interval with a one-sigma confidence interval[47] (See Supplementary Note 13); the horizontal error bars represent the error on the mean for $R_b$ measurements for the corresponding events. For the blue open circle in **b** as no 0-knot states are observed, we estimate $F_{2,tot}^{int}$ by finding the difference $F_{tot}(2, R_b) - F_{tot}(1, R_b)$ from experiment and extrapolating model predictions (red curve) to estimate the extra factor of $F_{tot}(1, R_b)$

knots states featuring many prime knots. This point can be made clear by quantifying the free energy of the knotting states.

Knot-formation is no longer kinetically limited at long-times where knot-formation probability asymptotes (Fig. 3b). In addition, the compressed chain is in a state of inhomogeneous equilibrium. Fluctuations of the chain can be analyzed via a generalized free energy change that is equivalent to the minimum work required to drive the system out of the inhomogeneous equilibrium state[27,31,32]. This generalized free energy change includes the change in equilibrium free energy plus work performed by external forces[27], work which in our case arises from the viscous force exerted by hydrodynamic flow on the knots.

We can use our knotting probability measurements to make estimates of the free energy changes associated with knot formation. Let $F_{tot}(m, R_b)$ be the total free energy change for forming a state with $m$ knots on a profile with barrier extension $R_b$ (in units of $k_B T$): $F_{tot}(m, R_b) = -\log Z(m, R_b)$ with $Z(m, R_b)$ the corresponding partition sum. The probability of forming $m$ knots is then

$$P(m, R_b) = Z(m, R_b) / \sum_{i=0}^{n_k} Z(i, R_b) \qquad (4)$$

Note that $Z(0, R_b) = 1$ as there is no free energy change for forming zero knots; $n_k$ is the maximum number of knots observed to occur (we find that $n_k = 3$). Using Eq. 4, we can directly extract knot formation free energies from experiment using $F_{tot}(m, R_b) = -\log Z(m, R_b) = -\log(P(m, R_b)/P(0, R_b))$. In addition, we introduce an interaction free energy for two knots $F_{2,tot}^{int}$. The interaction free energy gives the increased free energy of the two knot state over the free energy of the two knot state satisfying pure Poisson statistics. For example, if the two knots obey Poisson statistics, their partition sum $Z_P(2, R_b) = Z(1, R_b)^2/2!$, leading to $F_{2,tot}^{int}(R_b) \equiv F_{tot}(2, R_b) - 2F_{tot}(1, R_b) - \log(2)$.

can eliminate $n$ and express Eq. 2 purely in terms of the no-knotting ($m = 0$) probability $P_0$:

$$P_m = (-\log P_0)^m \frac{P_0}{m!}. \qquad (3)$$

Figure 5 shows Eq. 3 plotted for $m = 1$, 2, and 3 against the experimental data. Values of $P_0$ on the horizontal axis are calculated from the observed total knotting probability $P_0 = 1 - P_{total}$ for different values of $R_b$. Higher values of $P_0$ in Fig. 5 correspond to lower compression. The Poisson model describes the data well when the molecules are only slightly compressed and $P_0$ is close to unity. However, for high compression ($P_0 < 0.2$) the Poisson model breaks down. The breakdown in Poisson statistics can be explained in two ways: (1) at high compression, constituent prime knots might interact so that their formation is no longer independent; (2) at high-compression knots of complex topology are formed with higher probability, so that a single Poisson distribution does not reflect the overall knotting probability. We believe that knot interactions at high compression are the likely explanation, due to the absence of composite

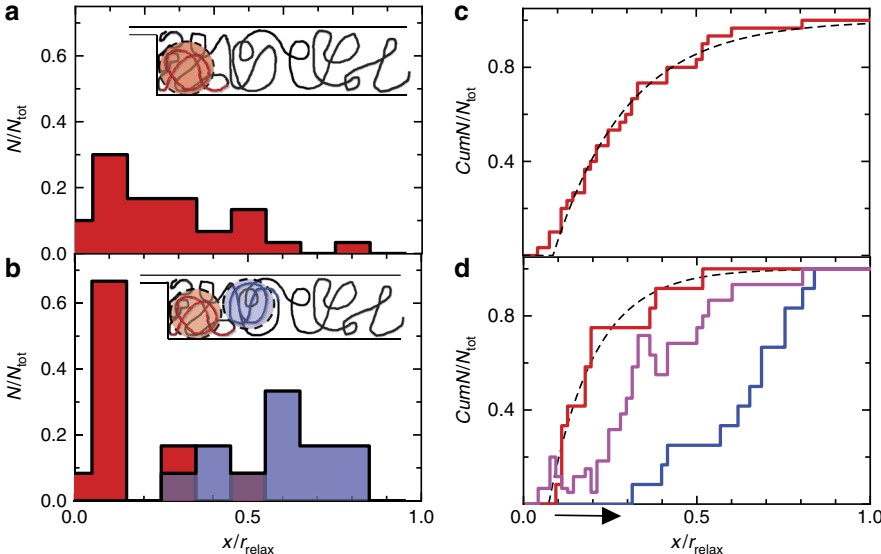

**Fig. 6** Knot position histograms. **a** Knot position histogram for one-knot states and **b** two-knot states 2 s after pressure release. **c** Cumulative knot position histogram for one-knot states and **d** two-knot states normalized to the total number of counts. The x-axis is normalized to the extension $r_{relax}$ of the relaxing molecule measured at the time for which the knot-position was obtained. The data used includes events with $R_b \approx 0.11$, 0.13, 0.17. For the two-knot states, we separately histogram the position of the knot closest to the barrier (lower knot, shown in red) and the knot farthest from the barrier (upper knot, shown in blue). The dashed curves in **c** and **d** are best-fits of respective data to the cumulative distribution corresponding to an exponential probability distribution. The arrow in **d** indicates that the upper knot distribution is shifted relative to the estimated non-interacting distribution (bold magenta)

Figure 4b gives the extracted free energies for forming a one knot state and knot interaction free energies as a function of $R_b$. The single-knot free energy becomes increasingly negative for small $R_b$. The interaction free energies are remarkably high (on order of several $k_B T$), suppressing multiple knot states. Knot–knot interactions, for example, could arise through the excluded volume of one knot restricting the configuration space of the other knots (knot–knot excluded volume); this effect would scale as $k_B T g_k^3/r D_1 D_2$ (with $g_k$ knot gyration radius). Yet, we expect the volume of a single knot to be very small relative to the volume occupied by the chain: with $g_k \sim 100\,\text{nm}^8$ we find $g_k^3/r D_1 D_2 \sim 10^{-2}$. The interactions must therefore have a more subtle origin.

**Knot spatial distribution**. Insight into the nature of the interactions can be gained by measuring the knot spatial distribution, which can be accessed a short-time following pressure release. Figure 6 shows the histogrammed position of knots for one-knot (Fig. 6a) and two-knot events (Fig. 6b), normalized to the chain extension $r_{relax}$, 2 s after pressure release. For the two-knot events, the position of the knot closest to the slit-barrier (lower-knot) and the knot farthest from the slit-barrier (upper knot) are separately histogrammed. In addition, we show the cumulative histograms for the single (Fig. 6c) and two knot case (Fig. 6d). As the cumulative histograms are insensitive to binning, we perform all quantitative analysis on the cumulative histograms.

The single-knot distribution (Fig. 6a, c) is non-uniform and well described by an exponential probability density function (Fig. 6c), suggesting that knots are found preferentially in concentrated regions of the chain. While we do not expect the probability distributions after release to quantitatively mirror the distributions for a compressed chain (there could be considerable complexity in how the evolving chain profile during relaxation affects the knot distribution), we can say that the distribution observed after pressure release represents a lower-limit on the degree of spatial non-uniformity present in the knot distribution prior to release (the relaxation process will smooth out an initially

non-uniform distribution but it will not introduce non-uniformity).

The two-knot spatial distributions have structure indicative of knot-interactions. Note that the upper knot distribution is shifted to larger $X$ relative to the lower knot distribution (as indicated by the arrow on the $X$-axis in Fig. 6d). If the two knots do not interact (i.e., so that they are statistically independent) and both satisfy a distribution peaked near the barrier (as observed for single knots) we would expect that both of the knots would be found with high probability near the barrier. Instead, there is gap in positions where only one knot is found. This gap could arise, for example, if the two knots interact like hard spheres over their gyration radius and satisfy a no-passing constraint, introducing a range in positions near the barrier where the second knot is physically excluded (see inset to Fig. 6b). Knot passing might be prohibited due to the large physical size of the knots, on order of the channel diameter[8,10], prohibiting knot crossing mechanisms that require knot expansion[33]. (Refer to Supplementary Note 5 for detailed discussion on knot interactions).

**Free energy model to quantify knot formation**. Our results suggest that knot-interactions are present at high compression, causing a pronounced deviation in observed knotting probability from Poisson statistics. The interactions may arise from a hard-core repulsion mechanism preventing knot crossing in the channel. Yet, it is unclear how the hard-core repulsion translates into a higher free energy cost for formation of multiple knots and leads to a breakdown in Poisson statistics. Here we develop one possible model to quantify the effect of a no-crossing constraint on the knot free energies, elucidate the role of compression in increasing knot-formation probability and explain the breakdown in Poisson statistics at high compression. Qualitatively, our model suggests that knot free energy is lowered during chain compression by an excluded-volume mechanism: knots tightly localize the contour they contain, avoiding the free energy cost that would be introduced by releasing the stored contour to interact with the rest of the compressed, concentrated chain. Moreover, our model

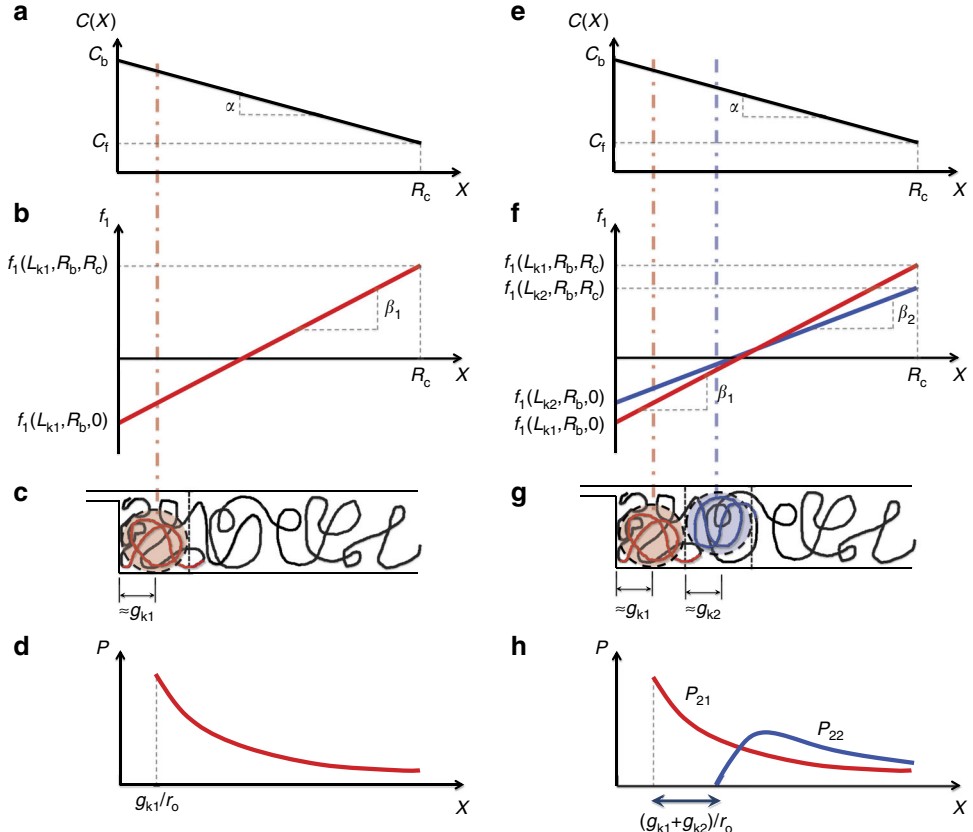

**Fig. 7** Free energy and probability distribution for one and two knot states. **a**, **e** Schematic of concentration profile $C(X)$, **b**, **f** resulting free energy profile $f_1(L_k, R_b, X)$, and **d**, **h** resulting knot probability distributions for single (**a**–**d**) and two-knot (**e**–**h**) states. Note that in the two-knot state the two knots can experience different free energy profiles due to their varying size (e.g., $L_{k1}$ does not necessarily equal $L_{k2}$): this is indicated in **f** by drawing two free energy profiles with slopes $\beta_1$ and $\beta_2$. In addition, in the two-knot state the knots will have different probability distributions: $P_{21}$ for the lower knot closest to the barrier and $P_{22}$ for the upper knot farthest from the barrier (see Supplementary Note 7 for detailed derivation of the probabilities) (**h**). **c** Cartoon of single-knot on chain, displaced by radius of gyration $g_{k1}$ from barrier edge. The dashed line relates the knot position to the corresponding concentration and free energy. **g** Cartoon of two knots on chain. Knot 1 (red) is displaced by radius of gyration $g_{k1}$ from barrier edge; knot 2 (blue), by assumption of single-file ordering, is displaced by $2g_{k1} + g_{k2}$ from barrier edge

suggests that the free energy of a knot should vary with position along the compressed profile, with the free energy lowest at the barrier-edge. If a no-crossing constraint exists, multiple prime knots cannot all occupy the position of minimum free energy, but instead will stack single-file, leading to an increased free energy of a composite knot state relative to the free energy of the independently formed prime knots.

Dai et al.[8] argue three types of free energy contribute to knot-formation: the energy of forming a knot on the chain in bulk $f_b(L_k)$, where $L_k$ denotes the contour length of the knot; the energy of confining the knot between the channel walls $f_{wk}(L_k, R)$ (wall knot) and the energy saved, as contour stored in the knot no longer contributes the confinement free energy $f_{wuk}(L_k, R)$ (wall unknot) associated with an unknotted section of polymer of contour $L_k$. In addition, the flow exerts a constant drag force $\zeta_k V$, leading to a free energy contribution $f_h \equiv \zeta_k V X / k_B T$ (in units of $k_B T$ with $\zeta_k$ a knot friction factor). The total free energy change $f_1(L_k, R_b, X)$ upon forming one knot of contour $L_k$ in the nanochannel at position $X$ on a profile with barrier extension $R_b$ is then

$$f_1 = A_b f_b + A_{wk} f_{wk} - f_{wuk} + A_h f_h \tag{5}$$

(See Supplementary Note 6 for explicit functional form). This equation makes explicit three dimensionless scaling constants $A_b$,

$A_{wk}$, and $A_h$ to be determined via least squares fitting. We expect trefoil knots to dominate[11]. Our model suggests that knot stabilization is driven by a large negative single-knot free energy at the slit barrier $f_1$, which varies linearly with position (Fig. 7b):

$$f_1(L_k, R_b, X) = f_1(L_k, R_b, 0) + \beta(L_k, R_b)X \tag{6}$$

(see Supplementary Note 7 for detailed derivation of equation 6 and explicit form of $\beta$ and $f_1(L_k, R_b, 0)$). If $f_{wuk}$ is large enough to ensure the free energy at the barrier $f_1(L_k, R_b, 0) < 0$, knots of size $L_k$ will exist with a spatial distribution $P_1(X) \sim \exp(-f_1(L_k, R_b, X))$ leading to an exponential accumulation near the barrier, consistent with the observed single knot distribution (Figs. 6a, c and 7d, Supplementary Note 8).

The barrier ($x = 0$), where the free energy is minimized, is the most probable location for a single knot to form, but knots can form at all x. Let the partition function $z_1(L_k, R_b)$ count the number of ways a single knot of size $L_k$ can form on a profile characterized by $R_b$. The number of statistically independent sites at which a knot can form is estimated by $n_{max} = r/2g_k$, each site $i$ weighted by a Boltzmann factor $\exp(-f_1(L_k, R_b, i2g_k/r_o))$, leading to a partition function that can be summed geometrically (see Supplementary Note 9). Similarly, the partition function of a state containing multiple ($m$) prime knots $z_m(L_{k1}\cdots L_{km}, R_b)$ is

calculated such that the linear ordering of the knots along the profile is preserved. At high compression, these partition functions contain only one state, a ground state configuration consisting of knots stacked single file, with no gaps, directly abutting the barrier (see Fig. 7c, g for one-knot and two-knot ground states, respectively). In this high compression limit Poisson statistics does not hold as only one state is accessible and strong interactions imply knots do not form independently. For low compression, knots can be excited away from the barrier. Interactions are weak as the knots are well separated and many states are accessible, leading to an emergence of Poisson statistics (see Supplementary Note 10).

Lastly, we must integrate over all knot sizes, forming a partition function:

$$Z(m, R_b) = (2P)^{-m} \int z_m(L_{k1} \cdots L_{km}, R_b) dL_{k1} \cdots dL_{km} \quad (7)$$

In practice, we obtain $Z(i, R_b)$ from direct numerical integration. Equations 4, 5, and 7 then enable determination of knotting probabilities as functions of $R_b$. Simultaneous least-square fitting of model predictions to the experimental one and two-knot formation probabilities (Fig. 4a) yields $A_b = 1.43 \pm 0.05$, $A_{wk} = 0.98 \pm 0.12$ and $A_h = 1.12 \pm 0.07$, on order of unity suggesting that the approach is self-consistent. Equivalently, we can fix $A_{wk} = 1$ and $A_h = 1$ and perform a one-parameter fit of the parameter $A_b$, which yields equivalent results (see Supplementary Note 11). Our theory captures the increasing knot formation probability with increasing compression, the non-monotonic behavior of the single knot formation probability (Fig. 4a) and the energy scales of knot interactions and single knot stabilization (Fig. 4b). Our model also predicts the very small number of observed three-knot events (Fig. 4a (inset)), a consequence of the large interactions. Finally, our model quantitatively captures the transition from Poisson statistics at weak compression to the non-Poisson regime at high compression (Fig. 5). The $A_b$ value required to get agreement with experiment is slightly higher than unity, possibly arising from physical effects, such as knot compression, that lead to higher knot free energy and are not included in the model.

## Discussion
In conclusion, we show that hydrodynamic compression induces DNA knotting in nanochannels with high probability. This is remarkable as it demonstrates that moderate confinement, two orders of magnitude weaker than that found in capsids, can also induce knot formation, suggesting a knot formation mechanism qualitatively different from what has been proposed in refs. [34,35] for capsids, where nematic ordering in strong spherical confinement can form toroidal knots with high probability. We show that the free energy scales for knotting under compression in the long-time limit can be estimated by extending known free energy scales for confined knots in a no-flow equilibrium. In addition, we find that knot-interactions likely exist, arising from a hard-core repulsion between knots preventing knot crossing and lead to single-file ordering of knots. Our model suggests knot interactions suppress multi-knot states and lead to a pronounced deviation from Poisson statistics expected for knot formation in a no-flow equilibrium limit.

In a recent study, Tang et al.[15] introduced a technique for inducing knots on DNA molecules via application of an AC electric field. From a practical point-of-view, our approach has the advantage that it is inherently parallel; many molecules can be simultaneously compressed in an array of nanochannels and their relaxed, nanochannel-extended states analyzed. From a physical

point of view, the approach of Tang et al. occurs in a much more complex, strongly non-equilibrium environment, with both solvent and DNA exhibiting complex dynamics resulting from the hydrodynamic instabilities induced by the DEP force. In particular, in the DEP approach the DNA tumbling dynamics leads to finite segmental current throughout the coil. In addition, the DNA is driven into a globule state, which is less well understood due to the complexity of the DEP-induced attractive interactions that drive the compression. In our system, based on geometric confinement and pressure-driven low Reynolds number flow, the DNA adopts a highly reproducible concentration profile that corresponds rigorously to an inhomogeneous equilibrium state with zero segmental current. The simplicity of the underlying DNA conformation in our system may facilitate modeling of knot generation processes. The transverse confinement in our system provides an additional parameter that can be used to tune knot-formation, with lower channel diameter in the extended de Gennes regime predicted to produce knots with greater probability at an equivalent degree of compression (see supplementary Note 12). In addition, the channel diameter likely sets an upper limit on knot size for channels below about 500 nm. Our approach may thus lead to composite knot states formed from smaller stacked prime knots distributed towards one molecule edge. In contrast, we speculate the approach of Tang et al. may lead to easier production of larger, topologically complex knots in the molecule center[13].

A complete understanding of knot formation in our system would require understanding the physics behind the lowered topological barriers leading to favorable kinetics at experimentally accessible time-scales. We do find that knotting probability rises with waiting time in the compressed state, with a kinetically limited regime at low waiting times. This appears to confirm the picture suggested by Meluzzi et al.[2] regarding the dependence of knotting probability on effective agitation time. In our microscopic experiment, for example, thermal fluctuations could supply the necessary agitation, or thermal fluctuations could be assisted by additional hydrodynamic effects. In their DEP-based compression experiments Tang et al.[15] hypothesize that a tumbling-like agitation is created by electric-field induced hydrodynamic instabilities.

In particular, lowering the topological barriers for knot-formation requires a mechanism for knot-ends to invade the main coil so that the chain ends can be threaded through internal loops in the chain[11]. In our experiment, one possibility is that the chain ends are forced in during the transient compression (shock-wave) process, although this does not explain the long observed waiting time. The second possibility is that subtleties of the steady-state hydrodynamic flow, perhaps curving streamlines near the slit barrier, might play a role in helping drive the chain ends into the coil. We feel, however, that this mechanism would need to be more subtle than the flow-induced tumbling described in ref. [15], as we expect the flow in our nanofluidic channels to be steady and laminar, leading to a static packing of DNA against the barrier rather than continuous recirculation or agitation. We do not apply an electric field, and we expect effects of electro-hydrodynamic coupling to be very weak, so there is no clear candidate for a physical effect that could create the recirculatory flow necessary to drive DNA tumbling. The third possibility is that thermal fluctuations alone are sufficient to drive the chain ends into the coil. At high compression linear ordering of blobs breaks down and the free energy barriers preventing long-range chain looping disappear[36]. Brownian dynamics simulations of our system[37] would help clarify which mechanism is correct. Yet, whatever physics drives the favorable kinetics, once we deduce that the kinetics are favorable by observing time-dependent saturation of knotting, our free energy approach is valuable as it

enables extraction of long-time knotting probabilities in a systematic way from knowledge of equilibrium behavior.

Our approach, like those explored in references[14,15], cannot form knots of known topology (in contrast to tweezers based approaches[4,22]). Directly tying knots with tweezers, however, is extremely challenging and low-throughput[22] and non-trivial to apply in confined systems. Lastly, while many experimental[13–15,22,38] and theoretical[8,20,21,39–42] studies agree that knots on chains are localized as tight knots, others[43–45] believe that knots can form which are not tightly localized and can spontaneously expand along chains. In our knot detection criteria, knots are persistent, localized and bright features representing metastable tight knots. While we do not observe diffuse knot configurations at no-flow equilibrium for long waiting times after chain relaxation, there might be some diffuse knotted configurations that we have missed in our knot numeration because they might unravel at short times during the relaxation process, especially if they are close to the molecule edges. Future Brownian dynamics simulations might elucidate the evolution of knots during molecule relaxation from the compressed state and estimate how many knots might be lost during this process.

In the future the knot factory could be further exploited to study the effect of channel width, ionic strength, DNA contour length and to generate knots for further dynamic studies. In particular, we expect the physics of knot formation to be very different in the transition ($D < 100$ nm) and Odijk confinement regimes ($D < 50$ nm) due to the qualitatively distinct underlying chain conformation in these regimes[28]. While we predict knotting probability increases with decreased channel width throughout the extended de Gennes regime, the situation for smaller channel width is unclear and a fascinating question for further theoretical and experimental study.

## Methods

**Device fabrication and experimental set-up.** The nanochannels are fabricated on fused silica substrates (HOYA) by electron beam lithography as described in ref.[28]. The slit barriers are formed by patterning the nanochannels with blunt ends in the array center (Fig. 1b). A 30-nm deep slit (measured using surface profilometry) is subsequently etched over the nanochannel array, transforming the blunt ends into barriers that will permit buffer flow but trap the DNA. In addition, adjoining the nanochannel array, the device contains two U-shaped microchannels (1 μm deep, 50 μm wide): these microchannels convey molecules from sand-blasted loading holes to the nanochannels. The $1 \times 1$ cm$^2$ chips are then bonded directly to fused silica coverslips (Valley Design; Fig. 1b). The cover slip seals the channels while the slit introduces an opening at the barrier end of the nanochannels, which allows flow to pass, but traps DNA molecules (Fig. 1c). Upon imaging the chip cross section using SEM, the nanochannels have horizontal dimension $D_1 = 325$ nm and vertical dimension $D_2 = 415$ nm (Supplementary Note 2 gives more detail on dimension acquisition). The loading buffer consists of 10 mM Tris titrated with HCl to pH 8.0. In addition, 2% $\beta$-mercaptoethanol (BME) is added to suppress photobleaching and photonicking. The DNA constructs used consist of T4 bacteriophage DNA (Nippon Gene, 166 kbp), stained with YOYO-1 (Life Technologies) at an intercalation ratio of 10:1, resulting in a contour length of about 63.7 μm[46]. The wet-chip is mounted on a chuck via o-ring seals with inlets for applying pneumatic pressure (Fig. 1a). The chuck-chip assembly is then mounted on an inverted microscope (Nikon Eclipse Ti-E) with a 100X N.A. 1.5 oil immersion objective. Imaging is performed via an EMCCD camera (ixon, Andor) with excitation illumination provided by a metal-halide lamp (Xcite). T4 DNA are driven into nanochannel arrays from loading microchannels via a burst of pneumatic pressure (Fig. 1b). Low ionic strength conditions (10 mM Tris, pH 8.0) are used to ensure negligible knotting probability in bulk by ensuring a high DNA effective width (Supplementary Note 2)[3]. The molecules are then driven to the array center, compressed against the slit-barriers to a well-defined extension $r$ and held for the waiting time $t_w$. After each compression event, molecules are driven out of the nanochannels, new molecules are introduced in order to avoid possible effects of entanglement[15] that might lead to hysteresis.

**Code availability.** The Matlab code used for image analysis in this study is available from the corresponding author upon reasonable request.

**Data availability.** The data that support the findings of this study are available from the corresponding author upon reasonable request.

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

## Acknowledgements

Funding is provided by Natural Sciences and Engineering Research Council of Canada (NSERC, Grant No. RGPIN 386212) and the Fonds de recherche du Québec-Nature et technologies (FRQNT, PR-180418). We thank LMN facility at INRS-Varennes, McGill Nanotools-Microfab, Facility for electron microscopy research (FEMR) and Prof. Coish, Prof. Flyvbjerg, Prof. Jona-Lasinio, and Prof. Landim for valuable discussion.

## Author contributions

W.R. and S.A. designed research; S.A. conducted experiments; A.K., P.Z., and S.A. designed, fabricated and characterized the device; S.A. performed data analysis; W.R., S. A., and L.Z. contributed in coding, S.A. and W.R. wrote the paper.

## Additional information

**Competing interests:** The authors declare no competing interests.

