## [Peer Review File · Nature Communications]

Reviewers' comments:

Reviewer #1 (Remarks to the Author):

The authors present experimental studies in which single DNA molecules become knotted after being confined in a nanochannel and compressed into the end of the channel by flow. These are nice experiments that seem to have been carefully done and several interesting conclusions are reached. The major findings are that the probability of the DNA forming a knot increases with increasing waiting time and with increasing amount of compression (tighter confinement). I am less qualified to comment on the theoretical models and I hope other reviewers will inspect those parts of the paper more closely. I feel that this paper may be appropriate for publication, and would potentially be of interest to a wide variety of researchers (ranging from mathematicians interested in knot theory, to polymer physicists, to molecular biologists), but I do have many questions and requests for clarifications that should first be addressed through revisions.

1. The abstract states "extension of the chain enables direct assessment of the polymer knotting state". This seems to really just mean "extension enables us to tell whether one, two, or three knots have formed in the polymer, provided they are sufficiently spatially separated". Many different kinds of knots exist (formally classified in mathematical knot theory) and saying "polymer knotting state" makes it sound like the technique can discern what types of knots form. But the technique can really only tell yes/no there is knot, so this should be clarified.

2. The abstract states "we propose that knotted states at high compression are stabilized as contour stored in the knot contributes a lower self-exclusion derived free energy". I was initially confused about what this meant and after reading the whole paper I have to admit I remained confused. First, I had trouble understanding what the experimental evidence is for knots being "stabilized", and what exactly is the meaning of the term "stabilized"? Is this necessarily implied by the experimental findings or rather something predicted by the theoretical model that is consistent with the experimental data, but not necessarily a unique possible model? As far as I understand, the experiment just shows that knots are formed and the probability that they are formed increases with increasing compression. How is this equated with "stabilization", which kind of implies that somehow the compression disfavors the knot becoming "untied", rather than just accelerating the formation of the knot in the first place? This needs to be clarified.

3. In the introduction, regarding the effect of confinement increasing the probability of knotting, wouldn't one expect that in the limit of very tight confinement that the probability of knotting would eventually decrease because the chain wouldn't be able to move around enough in a given incubation time to fluctuate into a knotted shape? For example, ZT Berendsen et al PNAS (2014) found that DNA confined in viral capsids can have a relaxation time longer than 10 minutes.

4. p. 1, line 55-57, for the experiments with agitated strings saying "saturated at the equilibrium value expected for a self-avoiding random walk", makes it sound like some a restricting limit was reached, but the experiments with strings, and theory for random walks, showed that probability of forming a knot approached 100% for long, flexible strings.

5. p. 1, line 75, "knots are present along the relaxed DNA, visualized as sharply localized regions of high intensity on the extended molecule." A skeptic could raise several criticisms here. First, if you can't trace the path of the chain how do you really know for sure that (topologically) you have a knot? It is possible that the chain segments are merely "bunched up" or "tangled" or in a "stuck unknot" (see: https://en.wikipedia.org/wiki/Stuck_unknot) or a "complex unknot" (see: <http://ashawfsi2013.weebly.com/introduction.html>) conformation such that there is locally a higher density of chain segments, but that topologically there is not actually a knot. It should be pointed out that this is a serious limitation/caveat of the present technique, as compared to, for example, the tying of knots with optical tweezers in Ref. 4, where the exact type of knot is known because its path was traced, or the study of knots in strings in Ref. 16, where the exact path of

the string was traced; (b) regarding detecting multiple knots, this would be limited by the spatial resolution of the imaging system. If two or more knots were too close together along the chain, they could not be resolved and would be mistaken for one knot.

6. I found the diagram of the fluidic channels (Fig. 1 a-e) confusing. I got the idea that the DNA goes down a thin channel and slams into the end, where the DNA is cannot move further due to its size but where water can flow through some small gap at the end. But I couldn't understand what the geometry actually is that allows the water to flow through – is it a small gap between the substrate and the coverslip at the end of the channel? Perhaps 2D diagrams with top and side views would be clearer?

7. Does a given flow rate always cause different single DNA molecules to undergo close to the same amount of compression, or is it variable for different molecules?

8. p. 2, As I understand, the authors propose a model which makes certain assumptions and they find that the model can predict some of the same trends observed experimentally. The question is how sure are we that the assumptions are obeyed and how many other models with different assumptions could have also possibly predicted those trends? I prefer wording at the beginning of the modeling section to the effect: "In this section, we propose one plausible theoretical model that can explain some of our findings". I urge caution with statements like p. 2, line 97 "This approach... is valid... if ...", because "is valid" can be read as implying that it has experimentally been proven valid, while I think the authors may just mean that some theoretical/mathematical assumption is theoretically valid in some limit. On Line 101, I prefer "this model suggests that..." to "our approach suggests...", to be clear that in this sentence they are referring mainly to the model.

9. Regarding the use of T4 DNA, a description should be given how this was obtained or prepared, and the length of the molecule stated. Also, a concern is that it is well known that long DNA molecules can easily be physically and/or chemically degraded during purification and/or handling. For example, the molecules can be broken through hydrodynamic forces when pipetting, or perhaps when flowed through a fluidic device (e.g. PF Davidson, PNAS (1959)). Or over time they can chemically degrade, or be degraded by stray nucleases. How do the authors know or ensure that the DNA in their experiments is not degraded? The probability to form a knot likely depends on the length of the DNA molecule and if the molecules are not all the same length this would affect the conclusions.

10. With regard to the time-evolution of the knotted state, the studies of knots with strings show that the probability to form a knot depends on the complexity of the knot. Therefore, when a chain is given more time to form a knot it not only does the probability of obtaining more knots increase, but the probability that the knot will attain a more complex form, that may have different properties than simpler knots, may change. It should be noted that since the authors cannot discern the complexity of the knots they cannot account for such effects.

11. With regard to plotting "Free Energy" as in Fig. 4, I feel it is important to clarify in the figure caption that this is really "Deduced free energy within the assumed model", not something that is directly experimentally measured.

12. Fig. 4 shows that the knotting probability is essentially zero when the compression is small. But isn't there supposed to be a finite probability that a long DNA molecule just floating around (unperturbed) has become knotted through thermal fluctuations (e.g., Shaw and Wang, Science (1993)). Is this just baseline level knotting negligible in the present case?

13. p. 5, line 238-241 "the compressed chain, possessing a time-invariant linear ramp profile, is in a state of mechanical and thermal equilibrium... we are thus justified in applying equilibrium statistical mechanics...". How do the authors know that the molecular conformation of the DNA

molecule is in the equilibrium state? Isn't it possible that the DNA is in a non-equilibrium conformation and very slowly relaxing (exploring different nonequilibrium conformations which may have similar ramp profiles), and/or perhaps only negligible relaxation occurs on the timescale of the measurement?

Reviewer #2 (Remarks to the Author):

The study of Amin et al. presents an interesting experimental study of the spontaneous knotting that occurs in 179-kb long DNA molecules that are compressed by a flow against a barrier.

I found the experimental results very appealing and insightful about how the metric and topological properties of the filaments are affected by spatial confinement. In addition, the setup allows to study various stages of the out-of-equilibrium response of DNA to compactification, which is a problem that is still largely unexplored despite its relevance for both applicative contexts and fundamental research.

For this reasons I believe the study contains a significant advancement to the phenomenological characterization of DNA behaviour. I have, however, some concerns about the interpretative framework that the authors used, especially regarding the analysis of knot-knot interactions and the application of equilibrium concepts in a steady-state situation where detailed balance should not apply.

I would be happy to recommend its publication in Nature Communications provided that the authors address these concerns, which are detailed below.

1. As I understand, establishing the lack of (significant) interactions between knots in equilibrated chains would require a different analysis from the one carried out here. One can see it rather clearly for model DNA chains inside ~ 100 -nm wide nano-channels where the knotting statistics has been shown to follow closely the Poissonian statistics for independent knotting events. However, if one was to compare the probability of having two knots $P(2)$ with the square of a single knot probability $P(1)*P(1)$, or equivalently compute $F(2) - 2 F(1)$, one would incorrectly conclude that knots interact with each other.

3. A further element that complicates the knot-knot interaction analysis is the non-uniform density profile of the DNA chain. In equilibrium knots occurrence rapidly increases with packing density. One could therefore imagine that one could more appropriately look at the incidence of one, two and more knots in various "longitudinal slices" of the system where the density is more uniform. Even if $P(2)$ was equal to $P(1)*P(1)$ in each slice, by the time it is integrated and normalised over the triangular density profile, the resulting $P(2)$ would not be equal to the square of the corresponding $P(1)$.

My personal take is that the steady-state condition realized by the authors is too complex to allow devising a transparent statistical model for establishing whether knots do or do not interact. However this, in my view, does not detract from the experimental study.

3. My other general concern regards the use of terms related to equilibrium thermodynamics to describe and interpret the system. Because the latter is maintained in out-of-equilibrium conditions, I think it would be best to refer to the long waiting time limit as a steady state situation rather than an equilibrium one.

In connection to this, I think that, differently from what stated at lines 53-57, previous studies on agitated strings (ref 2 and ref 83 therein) had not concluded that the observed knotting probability was equal to the equilibrium one of confined semiflexible chains. Those studies had, instead, reported that a kinetic model for the stochastic braiding of nearby strand could reproduce the increase of knots with agitation time.

4. The caption of Fig. 1, panel h, shows an example of a knotting event was discarded because the initially localised bright spot of the knot had become significantly more diffuse. This type of behaviour has, in fact, been reported in the spontaneous knotting and unknotting dynamics of flexible linear polymers.

So I wonder whether the conditions need to detect knots introduce a selection bias towards tight knots.

This would be a very important point to clarify for the benefit of e.g. future modelling studies seeking a quantitative agreement with the experimental measurements.

Reviewer #3 (Remarks to the Author):

The main aim of this paper is to design and implement a on-chip nanofluidic experiment that can produce, in a controlled way, (physical) knots in long linear DNA. The results presented clearly demonstrate that this can be done by first compressing the DNA, confined in a nanochannel, with a Poiseuille like flow and then letting the molecule to evolve into its relaxed confined configuration. The two main control parameters are the strength of the compressing flow, measured in terms of the barrier extension, and the amount of time during which compression is active or waiting time w_t . In particular, if w_t is sufficiently large, the DNA is shown to reach a steady state characterised by the formation of physical knots during the relaxation dynamics.

By varying, w_t the authors show that the probability of knot formation increases rapidly with w_t reaching a saturation value (plateau) close to one. This means that long DNAs, once compressed hydrodynamically within a nanochannel and let it relax, displays almost surely a knot.

Another interesting result is the condition under which the spontaneous formation of composite knots (made mostly by two prime knots) is observed: for very high compression (i.e. for sufficiently small values of the barrier extension) this probability becomes higher than the corresponding one for a single knot. This result suggests that if the control parameters are properly tuned, it is possible to increase the complexity of the formed knots, at least within the family of composite knots.

Finally it is of great interest the results on the spatial distribution of the formed knots: for a single prime knot it is shown that it preferentially forms and resides in the proximity of the barrier while, for a two-components composite knot, the distribution displays a non trivial behaviour suggesting an intriguing exclusion (topological) interaction between the two knots.

In recent years, there has been significant increase in the interest in knots in a range of different soft matter systems. This ranges from nematic liquid crystals to chemically synthesised knotted structures as well as to knotted polymers in Nature including proteins and DNA. There remain many fundamental issues about such structures including basic questions on their formation and also their dynamical behaviour. This paper should therefore be of wide interest to a number of different scientific disciplines and could attract the Nature Communications audience.

The manuscript has however some weak points that needs to be clarified/strengthened before being considered publishable in Nat Com. These points are listed below.

1) How the authors can be certain that the bright spots appearing during the relaxation dynamics are due to physical knots instead then, say, either to locally folded regions or to portions of the chain that are geometrically entangled but not necessarily knotted? One can imagine for instance to cases in which locally the chain seems to present a knotted portion but on larger scales it is actually unknotted. A typical example are slipknots that are known to occur very frequently in long chains and in proteins too. I think that the authors should make an effort to explain how they decide that the spots are indeed physical knots, i.e. portion of the chain that when properly closed

belong to a given knot type. I think that a deeper discussion on this issue is crucial to strengthen the validity of the results presented here.

2) Since this study presents a scheme of a knot making machine that is alternative to the one presented in Tang and co-workers (see Ref. 15), I think that a deeper comparison between the two schemes is needed to better emphasise the advantages/drawbacks of one approach with respect to the other.

In Tang et al the molecule considered is the same (T4) but the condensation of the molecule into small regions is due to the action of an external electric field. This field induces a collapse transition of the charged DNA into a globular, isotropic, conformation. Here, instead, the hydrodynamic compression gives rise to a rather anisotropic steady state configuration, as shown by the concentration profiles reported in Figure 1 (f-i). How this difference in the starting configuration impacts on the formation and complexity of knots during the relaxation? Which machine is in this respect more efficient? Which of the two protocols is more tunable and easier to control? How the knot spectrum depends on the two protocols? For example I suspect that, while in the isotropic configuration (i.e. induced by the electric field) the knots formed during the relaxation are essentially prime knots (although complex), the protocol introduced here should favour the formation of composite knots, as the results seem to suggest.

I think that the authors need to discuss these issues with some details, possibly in the concluding session.

3) One important result on the efficiency of the knot machine is the increase of the knotting probability with the increase of the waiting time t_w . This is clearly shown in Figure 3b where t_w is reported in seconds. In order to have a feeling of how large are these time intervals, I think it would be nice to compare them with some other known time scales for DNA molecules under confinement. For example for slit-like confinement it is known that the relaxation time of the T4 molecule within slits of widths $\sim 1/2$ microns is of the order of 2-3 seconds (see Balducci et al PRL 2007). If I am not mistaken there is also a paper, in which one of the authors is co-author, in which the relaxation dynamics of DNA molecules in channels has been discussed (It must be a PRL in 2005). I don't remember whether the DNA considered was a T4 or if some characteristic time scales have been explicitly reported. Nevertheless I think that a comparison between time scales is important. Note that in this case the DNA is in a compressed state and hence its relaxation time should be larger than the one expected for a DNA that has been simply inserted in a nanochannel.

4) How relevant is the degree of longitudinal confinement in the results presented here? Certainly the presence of the channel is crucial to establish the hydrodynamic laminar (very small Reynolds number) compressing field but is the confinement strong enough to affect also the configurational properties of the DNA in absence of the velocity field? Given the average extension in the bulk of a T4 molecule (~ 1.3 microns) and its effective diameter it is reasonable to expect that, for the chosen channel (with average width ~ 350 nm), the molecule in the relaxed (equilibrium) state is mildly confined i.e. at the beginning of the extended de-Gennes regime. Can the authors discuss a little bit more on this point?

5) Regarding the previous point I think that the issue of the effect that confinement can have on the efficiency of the knot making machine is a very important one and maybe some additional experimental tests or at least a deeper discussion could greatly improve and strengthen the importance of the results presented so far.

In this respect how difficult it would be to do some additional experiments in which either the width of the channel is reduced or, maybe simpler, a longer DNA molecule such as the lambda-DNA concatamers (the 6-lambda DNA used for example in Balducci et al PRL 2007) is used? Of course I am not asking to redo the full analysis reported so far in the paper but maybe a rough estimate of the knotting probability for a longer DNA (or smaller channel width), would be enough

to furnish some indication on how confinement can affect knot production. If, on the other hand, these additional tests would require too much time I still ask the authors to discuss this point in the text and provide an answer to this question with some arguments.

6) Perhaps the weakest point of the manuscript is the theoretical part where a free energy argument is used to get an estimate of the knotting probability as a function of the barrier extension (i.e. degree of compression). First of all I have found the assumption of thermal and mechanical equilibrium stated in line 240 a little bit arbitrary or at least not well explained. On which bases this assumption is made ? From a rigorous point of view the compressed states are not in equilibrium. One can however assume that these are steady states and propose a sort of generalised free energy argument for a steady states. Still the free energy that is considered has too many terms with free parameters that can be used to fit many possible scenarios. In this respect it is difficult to consider this theory a predictive one. Hence I suggest the authors either to add more convincing arguments on why one should believe on the predictive character of their theory or to put less emphasis on this part and saving more space for clarifying most of the unclear or poorly discussed issues mentioned above.

Summary of Changes to the Manuscript

We thank the reviewers for their constructive comments and criticisms. Here, we summarize the modifications we have made to the manuscript and to the supplementary materials:

1. *Knot-Formation Probability and Poisson Statistics*: Following the suggestions of the second reviewer, we compared our experimental data to the knot-formation model based on Poisson statistics introduced by Micheletti *et al.* [1]. Interestingly, the Poisson model explains our data at low compression (low pressure), while at high compression (high pressure), Poisson statistics fails to predict the knotting probability. The recovery of Poisson statistics in the low compression limit is very significant as it makes explicit the connection of our experiments to previous work on knot-formation in the no-flow equilibrium limit [1]. In order to include this discussion, we have modified the “*Knotting probability measurement*” of the manuscript with detail regarding Poisson statistics and added a new figure comparing our knotting probability measurements to the expected results for Poisson statistics (Figure 5 in manuscript). We have also highlighted our findings regarding Poisson statistics in the abstract and introduction.
2. *Generalized Partition Function and Poisson Statistics*: We believe now that a correct theory should yield Poisson statistics in the low compression limit where the chain approaches a no-flow equilibrium conformation (as suggested by the agreement of our knotting probability data with a simple Poisson model in the no-flow equilibrium limit, and the approach of Micheletti *et al.* [1]). While our original model is non-Poisson in *both* compression limits, in the revised version of the manuscript and supplementary materials we introduce a more general approach to the probability of knot formation on a compressed chain that *explicitly recovers Poisson statistics in the low compression limit*. Our generalized approach explicitly includes the contribution to the partition function of states where knots form away from the barrier (for single and multiple knot states). For high compression, the partition functions in the generalized approach (for single and multiple knot states) agree with our previous approach, with the partition functions dominated by a “ground state” consisting of knots stacked single file with no gaps directly abutting the barrier (in this limit the statistics are strongly non-Poisson). For low compression, where knots can form far away from the barrier and are widely spaced so interactions are weak, the generalized partition functions explicitly recover Poisson statistics. We think that the ability of our revised model to interpolate smoothly between Poisson and non-Poisson regimes (see Figure 5 in manuscript) adds significant impact. Due to the change in our model, we have modified the “*Knot formation free energy model*” section in the manuscript with detail regarding the calculation of generalized partition functions. We have also added sections VIII (“*Partition Functions for Knot Formation*”) and IX (“*Knot Statistics in the Low Compression Limit*”) to the supplementary file. Section VIII covers computation of the generalized partition functions and section IX discusses the low compression (Poisson) limit of these partition functions. The revised model leads to fitting parameters $A_b = 1.43 \pm 0.05$, $A_{wk} = 0.98 \pm 0.12$ and $A_h = 1.12 \pm 0.07$, which are close to the original values ($A_b = 1.17 \pm 0.12$, $A_{wk} = 1.1 \pm 0.25$ and $A_h = 1.27 \pm 0.1$).
3. *Knot Identification Criteria*: Following the comments of all the reviewers, we have added a description in the introduction section of the manuscript regarding criterion used to identify knots.
4. *Steady-state versus Equilibrium*: In response to questions from all three reviewers, we have clarified our language regarding what we mean when we say the profile is in equilibrium. Precisely, we mean that the profile is in a *zero-current steady-state*, which is equivalent to a state of *inhomogeneous equilibrium*. In particular, our answer to the second reviewer’s third comment provides a detailed discussion of this point.
5. *Comparison to DEP Knot Formation Approach*: In response to the third reviewer’s second comment, we have added a paragraph to the discussion section in the manuscript, comparing our knotting technique with the knotting method proposed in [2].
6. *Model Predictions for Varying Channel Diameter*: In response to the third reviewer’s fifth comment, we have added a section to supplementary materials (section XI, “*Model Predictions for Varying Channel Diameter*”) that gives our free energy model’s predictions for knotting probability as a function of channel diameter (at fixed degree of compression).
7. *Alternative Single Parameter Fit*: In response to the third reviewer’s seventh comment, we have added discussion of an alternative single-parameter fitting approach to supplementary materials (section X, “*Comparison to Experiment*”); this gives almost identical results to the original three-parameter fitting approach.

Some smaller changes include:

1. In response to the first reviewer’s fifth comment, we have compared our proposed knotting technique with the optical trapping approach in the manuscript.
2. In response to the first reviewer’s sixth comment, we updated the first figure in the manuscript to include a more detailed schematic of the nanochannels and barriers. We have also added a sentence in the methods section in order to clarify the structure of the nanochannels.
3. In response to the first reviewer’s 8th and 11th comments, we have fixed the respective sentences as well as the caption of figure 4 in the manuscript. Also, we added the contour length of stained T4 DNA in response to the first reviewer’s 9th comment.

There are also some minor changes to the manuscript that are discussed separately in the answer to each comment.

I. ANSWER TO REVIEWER 1

The authors present experimental studies in which single DNA molecules become knotted after being confined in a nanochannel and compressed into the end of the channel by flow. These are nice experiments that seem to have been carefully done and several interesting conclusions are reached. The major findings are that the probability of the DNA forming a knot increases with increasing waiting time and with increasing amount of compression (tighter confinement). I am less qualified to comment on the theoretical models and I hope other reviewers will inspect those parts of the paper more closely. I feel that this paper may be appropriate for publication, and would potentially be of interest to a wide variety of researchers (ranging from mathematicians interested in knot theory, to polymer physicists, to molecular biologists), but I do have many questions and requests for clarifications that should first be addressed through revisions.

We thank the reviewer for their positive comments.

1. The abstract states "extension of the chain enables direct assessment of the polymer knotting state". This seems to really just mean "extension enables us to tell whether one, two, or three knots have formed in the polymer, provided they are sufficiently spatially separated". Many different kinds of knots exist (formally classified in mathematical knot theory) and saying "polymer knotting state" makes it sound like the technique can discern what types of knots form. But the technique can really only tell yes/no there is knot, so this should be clarified.

Answer:

The reviewer makes a valid point. We can just count knots with our technique, not assess the topological state of the knots. Regarding the degree of spatial separation of the knots, we expect that even knots very close at one time will diffuse apart at later times, and then we could distinguish such knots if we measure long enough *as long as* the knots evolve independently (e.g. they are not entangled in such a way that they move together and cannot be resolved). We have changed the abstract to:

“Knots are produced during hydrodynamic compression of single DNA molecules against barriers in a nanochannel; subsequent extension of the chain enables *direct assessment of the number of independently evolving knots.*”

2. The abstract states "we propose that knotted states at high compression are stabilized as contour stored in the knot contributes a lower self-exclusion derived free energy". I was initially confused about what this meant and after reading the whole paper I have to admit I remained confused. First, I had trouble understanding what the experimental evidence is for knots being "stabilized", and what exactly is the meaning of the term "stabilized"? Is this necessarily implied by the experimental findings or rather something predicted by the theoretical model that is consistent with the experimental data, but not necessarily a unique possible model? As far as I understand, the experiment just shows that knots are formed and the probability that they are formed increases with increasing compression. How is this equated with "stabilization", which kind of implies that somehow the compression disfavors the knot becoming "untied", rather than just accelerating the formation of the knot in the first place? This needs to be clarified.

Answer:

Stabilization means only that the free energy decreases when knots form: this is an observation consistent with experiment, which shows a decreasing single knot free energy, and predicted by theory. Naturally, if the free energy decreases when knots form, then their untying will be disfavored. The intent of the language “knot stabilization,” as opposed to writing “knot formation” for example, is to provide a short-hand to distinguish between facts regarding the equilibrium state (which our model purports to describe) and the kinetic process whereby knots are formed (which our model does not describe). Note that equilibrium distributions are determined by the relative free energy changes

involved in forming the states in question; kinetics (e.g. whether transition rates are high or low) are determined by free energy barriers between states. Our model provides information only on the relative free energies of the states, not the nature of the barrier-crossing process itself, as is explained in the discussion. We think that the language “knot formation” implies understanding of both the kinetic and equilibrium aspects of the problem, while “knot stabilization” implies understanding of only equilibrium aspects, which is more limited and accurate. In order to clarify this point, we have modified the language in the abstract to:

“Our model suggests that knotted states at high compression are stabilized *by a decreased free energy* as contour stored in the knots. . .”

We have also tended to replace the language “knot stabilization” with the equivalent notion of “lowering knot free energy.”

3. *In the introduction, regarding the effect of confinement increasing the probability of knotting, wouldn't one expect that in the limit of very tight confinement that the probability of knotting would eventually decrease because the chain wouldn't be able move around enough in a given incubation time to fluctuate into a knotted shape? For example, ZT Berndsen et al PNAS (2014) found that DNA confined in viral capsids can have a relaxation time longer than 10 minutes.*

Answer:

We agree that jamming effects could be very relevant to knotting in tight confinement, but such effects lie outside the scope of the present study. Note that chains compressed in channels in an Odijk limit (persistence length P larger than channel width D , chain cannot backbend in channel) will behave very differently than chains in a de Gennes type limit (polymer can coil). Our model will without question breakdown as D approaches P . We know that knots do form in a channel close to P (around 60 nm [3]), but we don't have quantitative data to say whether the knotting probability is higher or lower for a given degree of compression. Whether the knotting probability will increase or decrease in a sub-persistence length channel, and how this would relate to jamming, is a fascinating question, but we do not believe the answer is obvious. For example, in an Odijk regime the compression mechanism will be quite different. We believe that a chain in an Odijk regime will compress through the nucleation of back-folds and the regular growth of internal loops [4]. This very organized internal structure could potentially reduce topological barriers for forming knots. There is also the question of how to understand knotting free energy in such small channels, as there is currently no known theory for the free energy of a compressed chain in these limits or knot confinement free energy.

To address this point we have added the following text to the conclusion: “ In particular, we expect the physics of knot formation to be very different in the transition ($D < 100$ nm) and Odijk confinement regimes ($D < 50$ nm) due to the qualitatively distinct underlying chain conformation in these regimes [5]. While we predict knotting probability increases with decreased channel width throughout the extended de Gennes regime, the situation for smaller channel width is unclear and a fascinating question for further theoretical and experimental study.”

4. *p. 1, line 55-57, for the experiments with agitated strings saying "saturated at the equilibrium value expected for a self-avoiding random walk", makes it sound like some a restricting limit was reached, but the experiments with strings, and theory for random walks, showed that probability of forming a knot approached 100% for long, flexible strings.*

Answer:

We agree that this statement is misleading and needs to be reworked. One issue is that, upon reading the two articles on this topic closely ([6] and [7] (see Figure 3)), we find that the discussion in each reference has a different emphasis based on the different (but related) results presented. For example, reference [6], which is based on tumbling experiments, suggests that the knotting probability should approach unity in the limit of high string length and flexibility (the highest measured knotting probability reported here, for strings of highest flexibility used, is 85%). Reference [7], in contrast, reports MD tumbling simulations (see Figure 3(f)) that clearly show for chains of finite size the knotting probability saturates at a value close to but below unity after several tumbles. Increasing the string length further in the simulations does not drive the knotting probability closer to unity (i.e. it stays at the saturating value). It does seem right to conclude, therefore, that the knotting probability saturates at a value below 100%, an effect probably related to finite string stiffness. There is an implication, although it is unproven, that making the strings even more flexible would drive the knotting probability even closer to 100%.

Upon reflection (and this is the major problem with the text as written in our view), it is unclear to us if the saturation value of knotting probability at long times really corresponds to the equilibrium knotting probability (i.e., as would be obtained from Monte Carlo simulations for the same system). We think that this *could* be the case, and the suggestion is certainly made in [6] that the knotting probability should approach 100% in the limit of a

long self-avoiding string, as this is the expected equilibrium result for a long self-avoiding string. However, this is an inference that goes beyond what is really demonstrated in the study. For example, the saturating value could result from a complex steady state that does not correspond exactly to equilibrium. Reference [6] presents a model for the knotting based on random braid moves between aligned regions of string that seems to capture many of the system’s features. This model is not quantitatively compared to the experiments, however, and it is unclear to us what long-time knotting probability it predicts (in particular, whether it can account for the saturating knotting probability).

Our final view is that the situation is unclear. We know for sure that a finite steady-state knotting probability is reached in the tumbling experiments after many tumbles; this probability can probably be pushed closer to 100% if the string is made longer and more flexible (and enough tumbles are performed). It is unknown if this steady-state probability represents the true equilibrium knotting probability of the confined chain, or would be better predicted by a kinetic model. We have revised the statement with wording that is more limited and accurate:

“At low agitation times, knot formation was observed to be kinetically limited; at longer agitation times, the knotting probability saturated at a value that approached unity for longer, highly flexible strings.”

5. p. 1, line 75, “knots are present along the relaxed DNA, visualized as sharply localized regions of high intensity on the extended molecule.” A skeptic could raise several criticisms here. First, if you can’t trace the path of the chain how do you really know for sure that (topologically) you have a knot? It is possible that the chain segments are merely “bunched up” or “tangled” or in a “stuck unknot” (see: https://en.wikipedia.org/wiki/Stuck_unknot) or a “complex unknot” (see: <http://ashawfsi2013.weebly.com/introduction.html>) conformation such that there is locally a higher density of chain segments, but that topologically there is not actually a knot. It should be pointed out that this is a serious limitation/caveat of the present technique, as compared to, for example, the tying of knots with optical tweezers in Ref. 4, where the exact type of knot is known because its path was traced, or the study of knots in strings in Ref. 16, where the exact path of the string was traced; (b) regarding detecting multiple knots, this would be limited by the spatial resolution of the imaging system. If two or more knots were too close together along the chain, they could not be resolved and would be mistaken for one knot.

Answer:

We divide the reviewer’s comment into four parts and answer each separately:

1. Knot Identification Criteria

While we cannot directly trace the chain path, there exist criteria that make “knots” on polymers distinguishable from “folds” or “trivial knots” (including tangled segments and complex unknots that do not possess true knot topology [2, 3, 8]). These criteria are motivated by A. Grosberg’s tight knot theory [9], now well-established by simulation [10, 11] and the experimentally observed behavior of objects with well-defined knot structure created via tweezers [8]. In Grosberg’s theory of tight-knot structure on semiflexible chains, a knot’s topological structure creates an effective network of non-crossing constraints that is equivalent to confining the chain in an effective tube. The knot free energy arises from two sources: (1) bending of the tube and (2) confinement free energy arising from polymer constrained in the tube. If the knot is small the bending energy is high as the tube is squeezed tight. Large knots have high confinement free energy as more polymer contour, pulled from less constrained regions of the chain, is forced to lie confined within the effective tube. The balance between bending and confinement free energy leads to a metastable knot size, giving rise to knots possessing a soliton-like structure [9] with a stable shape that diffuse along the chain through self-reptation of contour through the effective tube. In particular, in the Grosberg theory the metastable knot shape is stabilized by a high free energy barrier for introducing contour into the knot that prevents spontaneous knot loosening on experimentally accessible time scales (this barrier is estimated to be $\sim k_B T$ [9]). Knots tied by tweezers in fluorescently labeled chains, such as in [8], will adopt a bright (highly concentrated), localized (sub-diffraction) and stable structure that diffuses along the chain and *can only unravel at the chain ends*, consistent with the predictions of the Grosberg theory.

An entanglement in the contour is called a “knot” if it cannot be untied when the string is closed [7, 12]. In contrast, “trivial knots” or “unknots,” such as the “bunched up” or “tangled” regions the reviewer mentions, can unravel mid-chain (we discuss stuck unknots separately, see below). *Unknots* can be simple (in the form of \mathcal{S} or φ also known as “folds”) or complex, formed via concatenation of folds (“complex unknots”). Unknots, unlike knots, are not self-confined, thus they fluctuate in size significantly [3] and are eventually removed by thermal fluctuations. In particular, in a nanochannel, a concentrated region of DNA will have a higher free energy than the surrounding chain (due for example to self-exclusion interactions). Entropic forces will exist that will drive transport of contour from this concentrated region to non-concentrated regions, leading to large-scale unraveling dynamics (the simplest example is a confined polymer with a single-fold at the chain ends, which unravels via a simple kinetics described in [13]). If the structure possess true knot topology, this process will be halted

when the object reaches metastable knot-size, but unknots will continue unraveling until all contour is removed. Note that, for unknot structures, there is no built-in topology preventing the opening of tight loops by thermal fluctuations that rotate adjacent strands forming the loop. We thus expect small unknot structures to be highly unstable at the molecular level [7]; frictional effects do not play a role in stabilizing microscopic knots (unlike macroscopic knots), and these small unknot structures will possess high bending energy. Large, highly entangled structures may lead to arrested relaxation and take longer to unravel, as observed in [2], but our view is that only structures that possess a true knot topology will unravel to a final state possessing a diffraction limited “spot”-like structure.

We summarize our knot identification criteria as follows:

- *Knots are persistent, localized and bright features on extended chains*
- *Knots do not exhibit large-scale size fluctuations after reaching their final (metstable) state*
- *Knots unravel only at the molecule ends*

The above knot identification criteria have been used in similar experimental studies [2, 3] and have been followed rigorously in the current study. In the manuscript, we formalize our knot identification criteria and underlying rationale via the following text

“Knots can be distinguished from other topological events such as “folds” [13] or “trivial knots” (including entangled segments and complex unknots, like slip-knots [14], that do not possess true knot topology) as knots formed on semiflexible chains adopt a characteristic compact and time invariant structure [9–11]. A knot’s topology creates an effective network of non-crossing constraints that is equivalent to confining the chain in an effective tube. The balance between bending and confinement free energy of polymer in the tube leads to a metastable knot size with a high free energy barrier for knot loosening, giving rise to knots possessing a soliton-like structure [9] with a stable shape that diffuse along the chain through self-reptation of contour. For example, knots tied by tweezers in fluorescently labeled chains will adopt a bright (highly concentrated) and localized (sub-diffraction limit) structure that can only be removed when the knot diffuses to the chain ends [8]. In contrast, “trivial knots” or “unknots,” such as folds or entangled regions, are expected to gradually unravel under the influence of entropic forces driving contour to less confined regions [13]. Unknots can also unravel in mid-chain. Thus, in contrast to other topological events, knots are objects that once formed on the polymer: (1) are persistent, localized and bright features; (2) do not exhibit large-scale size fluctuations after reaching their final state; and (3) can unravel only at the molecule ends.”

2. Stuck Unknots

We believe that the mathematical concept of stuck unknot does not apply to semiflexible polymers (such as DNA) due to the prominent effect of thermal fluctuations at this scale, based on the following definition of stuck unknots:

The concept of “stuck (or locked) unknot” refers to **(1)** polygonal chains with **(2)** *fixed* edge length that **(3)** cannot be straightened (in open chains) or convexified (in closed chains) [15–17].

Note that the main feature of this class of structures is that *rigid* chain segments prevent continuous deformations of the contour [15, 18]. Thus, the determinative factor in introducing configurations which prevent disentanglement is *not* necessarily the topology, but the length of the rigid segments [18]. In this regard, even a trefoil knot on a chain with rigid segments can be made into a stuck unknot by fixing the lengths of the rigid segments to the appropriate values (Figure 1(a) in [18]). Thus, considering the semiflexible nature of DNA segments and their thermal fluctuations, the case of *stuck unknot* is unlikely to be found in DNA molecules.

3. Comparison to Optical Trapping Technique

As mentioned, using an optical trap to tie knots on polymers can create knots with a topology that is precisely known. The method employed in the current study and some previous ones [2, 3], on the other hand, don’t produce knots of known topology. The advantage of the current technique compared to optical trapping is its simplicity. As stated in [8], mechanical knotting of a molecule is challenging due to spontaneous relaxation of DNA molecule, requiring the use of special non-Newtonian solvents [19]. In the current study, on the other hand, we employ a simple compression technique, where all the parameters are under control. Also, the optical

trapping technique does not currently allow for production of knots in confined environments. We have added a sentence to the manuscript stating the limitation of the knot factory technique regarding determining knot topology:

“Our approach, like those explored in references [2, 3], cannot form knots of known topology (in contrast to tweezers based approaches [8, 19]). Directly tying knots with tweezers, however, is extremely challenging and low-throughput [8] and non-trivial to apply in confined systems.”

4. Detection of Multiple Knots

Note that in the current study an event with multiple knots is reported *only* when the knots diffuse apart so they can be separately identified. Figure 1(i, j) in manuscript gives an example of a two-knot and three-knot event where knot diffusion separates the knots on the chain so they can be distinguished.

6. *I found the diagram of the fluidic channels (Fig. 1 a-e) confusing. I got the idea that the DNA goes down a thin channel and slams into the end, where the DNA is cannot move further due to its size but where water can flow through some small gap at the end. But I couldn't understand what the geometry actually is that allows the water to flow through is it a small gap between the substrate and the coverslip at the end of the channel? Perhaps 2D diagrams with top and side views would be clearer?*

Answer:

We understand that the diagrams could be confusing, thus we have added a panel to Figure 1 (Figure 1(c)), illustrating the cross section of a nanochannel. Moreover, we have added a new line to the manuscript under “Methods” section, which explains the structure of the microfluidic chip. The added line reads as follows:

“... (Figure 1(b)). The cover slip seals the channels while the slit introduces an opening at the barrier end of the nanochannels, which allows flow to pass, but traps DNA molecules (Figure 1(c)).”

7. *Does a given flow rate always cause different single DNA molecules to undergo close to the same amount of compression, or is it variable for different molecules?*

Answer:

Different molecules of the same size compressed in the same channel dimension give a comparable degree of compression. We present results in the manuscript concerning this question. Figure 2(g) in the manuscript gives results for the single molecule extension r normalized to the equilibrium extension r_o (i.e. the quantity R_c) as a function of V . Each one of the points on the plot represents a measurement for a single molecule, so the degree of scatter in densely sampled regions of the plot (consider the region at around $V = 10 \text{ m}\mu/\text{s}$) gives the variability in the observed compression over different molecules. We can also average the data in certain regions of the plot and report the degree of scatter as a standard-deviation on the resulting R_c value. The flow velocities $V \simeq 1.46 \pm 0.08 \mu\text{m}/\text{sec}$, $V \simeq 10.78 \pm 0.21 \mu\text{m}/\text{sec}$ and $V \simeq 20.64 \pm 1.17 \mu\text{m}/\text{sec}$ result in $R_c = 0.52 \pm 0.02 \mu\text{m}$, $R_c = 0.198 \pm 0.003 \mu\text{m}$ and $R_c = 0.127 \pm 0.004 \mu\text{m}$ (each point is respectively averaged over 8 – 13 values). The reported standard deviations on R_c resulting from measurements over the different molecules are below 5%.

8. *p. 2, As I understand, the authors propose a model which makes certain assumptions and they find that the model can predict some of the same trends observed experimentally. The question is how sure are we that the assumptions are obeyed and how many other models with different assumptions could have also possibly predicted those trends? I prefer wording at the beginning of the modeling section to the effect: "In this section, we propose one plausible theoretical model that can explain some of our findings". I urge caution with statements like p. 2, line 97 "This approach is valid if ", because "is valid" can be read as implying that it has experimentally been proven valid, while I think the authors may just mean that some theoretical/mathematical assumption is theoretically valid in some limit. On Line 101, I prefer "this model suggests that" to "our approach suggests", to be clear that in this sentence they are referring mainly to the model.*

Answer:

We agree with the reviewer. While the proposed model predicts the molecule behavior under certain assumptions, it does not rule out the possibility of other models capable of explaining the observed behavior. We have added the following sentence to the beginning of the free energy model section: “Here we develop one possible model to quantify the effect of a no-crossing constraint on the knot free energies, elucidate the role of compression in increasing knot-formation probability and explain the breakdown in Poisson statistics at high compression.” We are now more careful with wording, making clear a difference between a fact the “model suggests” and what “experiment suggests.” “This approach ... is valid ... if ...” has now changed to “This approach ... explains our observations if ...”.

“our approach suggests ...” has been also replaced by “our model suggests that ...”.

9. Regarding the use of T4 DNA, a description should be given how this was obtained or prepared, and the length of the molecule stated. Also, a concern is that it is well known that long DNA molecules can easily be physically and/or chemically degraded during purification and/or handling. For example, the molecules can be broken through hydrodynamic forces when pipetting, or perhaps when flowed through a fluidic device (e.g. PF Davidson, PNAS (1959)). Or over time they can chemically degrade, or be degraded by stray nucleases. How do the authors know or ensure that the DNA in their experiments is not degraded? The probability to form a knot likely depends on the length of the DNA molecule and if the molecules are not all the same length this would affect the conclusions.

Answer:

We have added a description of T4 DNA preparation, including the length of the stained DNA, to the “Methods” section:

“...stained with YOYO-1 at an intercalation ratio of 10:1, resulting in a contour length of about $63.7\mu\text{m}$ [20].”

T4 DNA is considered a relatively long DNA, thus it is susceptible to breakage during handling/purification. The T4 DNA molecules used in our experiments have already been purified by the provider (Nippon Gene company) and guaranteed to be intact (166kbp). Special care is made while pipetting/injecting the DNA solution to induce minimal agitation of the molecules and the reducing agent β -mercaptoethanol (BME) is added to avoid photonicing, but some fragmentation is inevitable. There are several studies that have assessed the extension of T4 DNA as a function of channel size and/or buffer salt concentration [21–23], while others [24, 25] have used the extension of DNA molecule at equilibrium to distinguish intact from fragmented T4 DNA. Figure 1 illustrates the distribution of T4 DNA molecules’ extension in our nanochannels at equilibrium. The extension values are spread around the mean value $14.3\mu\text{m}$, with relatively small error on the mean ($0.3\mu\text{m}$).

FIG. 1. Histogram of the distribution of observed T4 DNA extension in nanochannel. The graph depicts the percentage of the molecules (vertical axis) found at equilibrium extension (horizontal axis). Taking the average of the observed extensions and calculating the error on the mean, we approximate the extensions as $14.3 \pm 0.3\mu\text{m}$.

10. With regard to the time-evolution of the knotted state, the studies of knots with strings show that the probability to form a knot depends on the complexity of the knot. Therefore, when a chain is given more time to form a knot it not only does the probability of obtaining more knots increase, but the probability that the knot will attain a more complex form, that may have different properties than simpler knots, may change. It should be noted that since the authors cannot discern the complexity of the knots they cannot account for such effects

Answer:

We agree with the reviewer that we do not access knot topology and cannot therefore account for effects associated with knot topology. Certainly, the kinetic barriers for forming topologically complex knots are probably higher than simple knots, so it is conceivable the system could reach an equilibrium with respect to production of topologically

simpler knots at shorter times. On the other hand, while our technique cannot directly access knot topology, it is possible that in the future studying multi-step knot decays, such as explored in [26], might lead to information regarding knot topology (at least to the degree to be able to tell trefoil knots apart from highly complex knots). Thus, we address this comment by adding to the conclusion:

“Lastly, while our approach, like that of Tang *et al.*, can not directly access knot topology, recent work suggests that knot complexity can be accessed by the presence of multi-step knot decays as knots reach the chain ends and unravel sequentially [26]. In a nanochannel, the molecule extension does not decrease as a knot approaches the molecule edge, possibly leading to enhanced resolution of multi-step decays that would add significant topological information to future knot-formation studies, clarifying how knot complexity might evolve with waiting time and degree of compression.”

We also explicitly state that our method cannot access knot topology in the introduction section, when we compare our approach to the method of knot-formation by direct tying in an optical trap.

11. *With regard to plotting "Free Energy" as in Fig. 4, I feel it is important to clarify in the figure caption that this is really "Deduced free energy within the assumed model", not something that is directly experimentally measured.*

Answer:

We partially agree with the reviewer. These values are taken directly from experiment and don't rely on any detailed assumptions of our free energy model, but they do require that the profile is in a state of inhomogeneous equilibrium (e.g. so that a free energy can be meaningfully defined). We have written: “Free energy of single knot states $F_{\text{tot}}(1, R_b)$ (red, circles) and two-knot interaction free energy $F_{2,\text{tot}}^{\text{int}}(R_b)$ (blue, circles) *deduced assuming the profile is in a state of inhomogeneous equilibrium*, with theoretical overlay using same fitting parameters for (a).” (italics represents added text).

12. *Fig. 4 shows that the knotting probability is essentially zero when the compression is small. But isn't there supposed to be a finite probability that a long DNA molecule just floating around (unperturbed) has become knotted through thermal fluctuations (e.g., Shaw and Wang, Science (1993)). Is this just baseline level knotting negligible in the present case?*

Answer:

The experimentally observed baseline knotting level is negligible. Note that we are working with a low salt buffer (10 mM) tris which gives rise to a high DNA effective width $w = 17$ nm (the effective width is a measure of the range of repulsive electrostatic interactions). A high effective width is known to suppress knotting in bulk [27]. Note that simulations investigating knotting along nanochannel extended chains, which predict a high knotting probability, use low effective widths ($= 2$ nm). Moreover, if knots could form on nanochannel extended chains with moderate or even low probability we expect that we would have seen them, yet we have never observed knots to form spontaneously on nanochannel extended DNA molecules (working in buffers < 100 mM). While we do not expect the equilibrium knotting probability to be exactly zero, even for a high effective width [11], there are additional *kinetic* limitations preventing formation of knots along a nanochannel confined chain in the absence of flow. For example, the free energy cost for a back-fold to form is high in a nanochannel; it is possible that this makes knot-formation along an extended chain highly unfavorable (i.e. give rise to a high free energy barrier for knotting for an equilibrium extended chain, even if the equilibrium knotting probability is not zero). Of course, this barrier will also be a strong function of the effective width, so it is possible that knots could form spontaneously in high salt conditions.

13. *p. 5, line 238-241 "the compressed chain, possessing a time-invariant linear ramp profile, is in a state of mechanical and thermal equilibrium we are thus justified in applying equilibrium statistical mechanics". How do the authors know that the molecular conformation of the DNA molecule is in the equilibrium state? Isn't it possible that the DNA is in a non-equilibrium conformation and very slowly relaxing (exploring different nonequilibrium conformations which may have similar ramp profiles), and/or perhaps only negligible relaxation occurs on the timescale of the measurement?*

Answer:

There are two parts to our answer. In part 1, we explain why we believe we are observing a steady-state on experimental time-scales. (We think this information is important as it makes clear that we don't have experimental evidence for relaxation processes the reviewer mentions, but it won't answer the bulk of the reviewer's question). In part 2, we explain why the steady-state is equivalent to a state of equilibrium (more precisely, “inhomogeneous equilibrium”).

Experimentally, we observe that once the transient compression phase is over, the compressed DNA undergoes thermal fluctuations about a well-defined profile shape. In particular, Fig. 2 shows the normalized extension (Fig. 2(a,b)) and ramp-slope (Fig. 2(c,d)) versus time of DNA molecules compressed via two different flow velocities (for the steady-state portion of the compression, i.e following completion of the transient). Evidently, the molecule extension in the compressed state and the profile slope appear to fluctuate around a fixed average. We see no evidence of a slow-relaxation over the time-scales of the experiment (i.e. the data in both cases is flat).

For the second part of our answer, which is also raised by the second reviewer, we refer to our reply to reviewer 2's third question. Briefly, as the reviewers have pointed out, the fact that we reach a steady-state does not mean that the steady-state is an equilibrium state. We believe that the observed steady-state is in fact an equilibrium state (or more precisely a state of "inhomogeneous equilibrium"), because it has quantitative characteristics we would expect if it is determined by a local mechanical equilibrium between the osmotic pressure gradient and applied hydrodynamic force, leading to zero segmental current J everywhere along the profile, or a zero segmental drift velocity (by the word "segment" we refer to Kuhn segments making up the DNA). The key points of agreement are that we observe: (1) a linear ramp and (2) quantitative scalings for the compressed extension and ramp slope as a function of V that are consistent with equilibrium predictions (see Fig. 2(g-j) in the manuscript). This type of zero-current state resulting from a local force balance is equivalent to an inhomogeneous equilibrium, or force constrained equilibrium and arises in many soft-matter systems subjected to constant forcing against a hard barrier [28]. Note that when we use the word equilibrium, we are referring to the DNA molecule alone. The solvent of course is not in equilibrium as the solvent is undergoing steady ultra low Reynolds number flow, constantly penetrating the molecule, but the effect of the steady-solvent flow on the DNA is simply to supply a source of static forcing.

FIG. 2. Measurements of normalized extension and slope of intensity profiles at compressed state for two different flow velocities $V = 9.9\mu m/sec$ (left) and $V = 12.8\mu m/sec$ (right). (a,b) Extension of DNA molecules at compressed state normalized to the initial extension of the molecules. The solid black lines depict the average extensions $r_c = 0.21 \pm 0.02$ (a) and $r_c = 0.18 \pm 0.01$ (b). (c,d) The slope of intensity profiles at compressed state. The solid black lines show the average slopes $\alpha = 30.2 \pm 3.7$ (c) and $\alpha = 39.25 \pm 4.6$ (d).

II. ANSWER TO REVIEWER 2

The study of Amin et al. presents an interesting experimental study of the spontaneous knotting that occurs in 179-kb long DNA molecules that are compressed by a flow against a barrier. I found the experimental results very appealing and insightful about how the metric and topological properties of the filaments are affected by spatial confinement. In addition, the setup allows to study various stages of the out-of-equilibrium response of DNA to compactification, which is a problem that is still largely unexplored despite its relevance for both applicative contexts and fundamental research.

For this reasons I believe the study contains a significant advancement to the phenomenological characterization of DNA behaviour.

We thank the reviewer for their positive comments. We also note the reviewer's interest in out-of-equilibrium aspects of the experiment, which we appreciate.

I have, however, some concerns about the interpretative framework that the authors used, especially regarding the analysys of knot-knot interactions and the application of equilibrium concepts in a steady-state situation where detailed balance should not apply.

While we agree that caution is required in applying equilibrium concepts to steady-state situations, in this case we believe that we are justified in using equilibrium concepts due to the fact that we are dealing with a very special type of steady-state: a steady state with a zero current (of DNA segments). Such a steady-state can be mapped to a state of inhomogeneous or force-constrained equilibrium. We have expanded on this point in detail in answer to the reviewer's comment 3.

*1. As I understand, establishing the lack of (significant) interactions between knots in equilibrated chains would require a different analysis from the one carried out here. One can see it rather clearly for model DNA chains inside 100-nm wide nano-channels where the knotting statistics has been shown to follow closely the Poissonian statistics for independent knotting events. However, if one was to compare the probability of having two knots $P(2)$ with the square of a single knot probability $P(1)*P(1)$, or equivalently compute $F(2) - 2 F(1)$, one would incorrectly conclude that knots interact with each other.*

*2. A further element that complicates the knot-knot interaction analysis is the non-uniform density profile of the DNA chain. In equilibrium knots occurrence rapidly increases with packing density. One could therefore imagine that one could more appropriately look at the incidence of one, two and more knots in various "longitudinal slices" of the system where the density is more uniform. Even if $P(2)$ was equal to $P(1)*P(1)$ in each slice, by the time it is integrated and normalized over the triangular density profile, the resulting $P(2)$ would not be equal to the square of the corresponding $P(1)$. My personal take is that the steady-state condition realized by the authors is too complex to allow devising a transparent statistical model for establishing whether knots do or do not interact. However this, in my view, does not detract from the experimental study.*

Answer:

These two comments, which are best answered together, raise an extremely interesting point and have led (we believe) to a considerable improvement in our manuscript, both in the interpretation of the data and the theoretical model (summarized in Fig. 5 in the manuscript).

We agree with the reviewer that a chain with independent knotting events should display Poisson statistics. In particular, for a chain in no-flow equilibrium such as studied by Micheletti *et al.*, the Poisson model suggests that the probability of forming a composite knot based on m number of prime knots of the same topology is governed by

$$P_m = n^m \frac{e^{-n}}{m!} \quad (1)$$

with $n = L/L_0$ where L is the DNA contour length and L_0 a characteristic length scale depending on the channel width D . While the concentration profile is uniform for a chain in no-flow equilibrium, note that concentration uniformity is *not* required for Poisson statistics to hold; Poisson statistics requires only that the prime knots form *independently*. In an inhomogeneous Poisson process, the event probability (in our case the knot formation probability) can vary along the chain, leading to a distribution identical to Eq. 1 but with n expressed as an integral of the varying knot formation probability along the chain [29]. For both the uniform and non-uniform cases, we can eliminate n and express Eq. 1 purely in terms of the no-knotting ($m = 0$) probability P_0 :

$$P_m = (-\log P_0)^m \frac{P_0}{m!}. \quad (2)$$

Thus, in the absence of interactions, and if knots have the same topology, Eq. 2 should describe knotting statistics on both uniform and compressed chains! We compared Eq. 2 to our experimental results (Fig. 5 in manuscript) and found that Poisson statistics worked well at low compression but failed at high compression. While the failure of Poisson statistics might result from inclusion of knots of complex topology, we think knot interactions are the more likely explanation as composite knot states with a large number of knots are strongly suppressed. In addition, an interaction hypothesis is consistent with the observation that the probability measurements in the low compression limit agree with Poisson statistics (where knots are distributed across the chain and expected to interact weakly), whereas deviations occur in the high compression limit (where knots are forced against the barrier and expected

to interact strongly). This argument is included in a revised section “*Knotting probability measurement*” in the manuscript and a new figure that shows the comparison to Poisson statistics (Fig. 5 in the manuscript).

Regarding the free energy measurements, we believe that we simply need to slightly redefine the interaction free energy. The important point is that Poisson statistics establishes the true baseline that would be satisfied by non-interacting knots. In particular, the interaction free energy for two knots $F_{2,\text{tot}}^{\text{int}}$ should give the increased free energy of the two-knot state over the free energy of the two-knot state satisfying pure (inhomogeneous) Poisson statistics. For example, if the two knots obey Poisson statistics, their partition sum $Z_P(2, R_b) = Z(1, R_b)^2/2!$, leading to $F_{2,\text{tot}}^{\text{int}}(R_b) \equiv F_{\text{tot}}(2, R_b) - 2F_{\text{tot}}(1, R_b) - \log(2)$. Thus, the only difference from before is the extra factor of $\log 2 \approx 0.7$, which only slightly lowers the interaction free energy and does not affect the results.

In addition, inspired by this discussion, we were able to generalize our free energy model to give results consistent with Poisson statistics in the low compression limit and describe the departure from Poisson statistics in the strong compression limit.

3. *My other general concern regards the use of terms related to equilibrium thermodynamics to describe and interpret the system. Because the latter is maintained in out-of-equilibrium conditions, I think it would be best to refer to the long waiting time limit as a steady state situation rather than an equilibrium one.*

Answer:

We agree that this point needs to be clarified. Our system is maintained in a *zero-current steady state*: this is equivalent to a state of *inhomogeneous equilibrium*, or *force-constrained equilibrium* where local forces everywhere are in balance. As has been rigorously demonstrated for a broad class of systems, including systems described by underlying hydrodynamic/diffusive equations (such as the nonlinear convective-diffusion formalism we use to describe the compressed DNA) [28, 30], fluctuations from such steady-states can be analyzed via a generalized free energy change that is equivalent to the minimum work required to drive the system from the state of inhomogeneous equilibrium [31, 32]. This generalized free energy includes the change in equilibrium free energy (work in absence of external forcing) plus work performed by external forces.

While flow is constantly penetrating the DNA coil and passing through the slit (in this sense the system is out-of-equilibrium), this flow is steady and necessarily laminar due to the ultra low Reynolds number of the nanochannels. Such a flow-field creates a source of *static external forcing* on the DNA, driving our chain against the barrier. Our problem is thus analogous to a chain pushed against a barrier by a static force, and bears resemblance to other problems in soft-matter where a system is driven by a force against a hard-wall, such as sedimentation, or centrifugation. Critically, the long-time-limit of such a problem corresponds to a special type of steady-state where the current J (that is current of DNA Kuhn segments in our case) is zero. In these systems the current vanishes even in the presence of a non-uniform concentration profile. A classic example would be the barometric distribution of concentration in a constant gravitational field. In general, the vanishing of the current gives rise to an ODE that describes the steady-state profile. In the context of the nonlinear convective-diffusion formalism we use to describe the compressed DNA (see supplementary materials section II(D)), the zero-current condition gives rise to a linear ramp (Eq. 14 in II(D) is the ODE that follows from setting $J = 0$; Eq. 14 yields a linear concentration ramp, Eq. 15). Note we have strong *experimental* evidence that the compressed DNA does indeed correspond to such a zero current steady-state: (1) the DNA concentration profile has the shape of a linear ramp and (2) quantitative scalings for the compressed extension and ramp slope as a function of V are also consistent with a zero current steady-state (see Fig. 2(g-j) in manuscript). This experimental evidence suggests that physical effects that would violate a simple zero current condition (e.g. recirculatory flows) do not exist, or at least do not occur at scales that we can resolve.

A “zero current steady-state” is equivalent to an “inhomogeneous equilibrium” or a “force-constrained equilibrium,” where external forces (in our case hydrodynamic) everywhere balance osmotic pressure ([28], note that the language ‘inhomogeneous equilibrium’ is standard language in the field of non-equilibrium thermodynamics and is taken directly from reference [28]). Locally, states of inhomogeneous equilibrium look like states of homogeneous equilibrium. In particular, if an inhomogeneous equilibrium state is examined in a large enough microscopic neighborhood, it looks like a homogeneous equilibrium state chosen with the concentration value corresponding to a particular point on the inhomogeneous profile. For a state of inhomogeneous equilibrium, the concentration profile can also be obtained from the variational derivative of a free energy incorporating a term describing the work performed by the external forcing. Fluctuations away from a state of inhomogeneous equilibrium can be described by probabilities proportional to $\exp(-\Delta F/k_B T)$ with ΔF a corresponding generalized free energy change related to the minimum work required to drive the system from the state of inhomogeneous equilibrium [28]. This generalized free energy includes the change in equilibrium free energy appropriate for the changing concentrations arising from the fluctuation plus work performed by the external forces (see [31, 32] for examples of related work using this approach to analyze fluctuations away from constrained polymer steady-states in nanofluidic systems). We argue that the probability of forming a knot at a particular position x on the profile, with the knot viewed as a small excitation away from the state of inhomogeneous

equilibrium, can be characterized by $\exp(-\Delta F_k(x)/k_B T)$ with ΔF_k the generalized free energy change including the change in equilibrium free energy arising from knot formation at x plus the work performed by hydrodynamic forces on the knot (as discussed in detail in our manuscript).

To clarify this point, we make the following change in wording. Original text reads: “To rationalize our findings, we argue that the compressed chain is in a force-constrained equilibrium at long times, satisfying thermal and mechanical equilibrium, so that statistical mechanics can be applied to quantify knotting probability and extract free energy scales for knotted states” \Rightarrow Modified text: “To rationalize our findings, we argue that the compressed chain is in a steady-state with zero segmental current, equivalent to a state of inhomogeneous equilibrium, so that a generalized free energy can be developed to quantify the probability of knot-formation.”

We have also added the following text in the section “DNA concentration profile during compression:” “The second phase begins when the shock-wave reaches the free edge. In this second phase, the laminar flow forces the chain immobile against the slit barrier with forces due to the osmotic pressure gradient everywhere balancing hydrodynamic forces so that the net polymer current is zero (e.g. zero net movement of Kuhn segments). *This zero current steady-state is equivalent to a state of inhomogeneous or force-constrained equilibrium [28].*” (italics represents added text)

In the section “Knotting probability measurement,” before we describe how we extract the free energies from the experimental data, we add the text: “Knot-formation is no longer kinetically limited at long-times where knot-formation probability asymptotes (Fig. 3(b)). *In addition, the compressed chain is in a state of inhomogeneous equilibrium. Fluctuations of the chain can be analyzed via a generalized free energy change that is equivalent to the minimum work required to drive the system out of the inhomogeneous equilibrium state [28, 31, 32]. This generalized free energy change includes the change in equilibrium free energy plus work performed by external forces [28], work which in our case arises from the viscous force exerted by hydrodynamic flow on the knots.*” (italics represents added text)

In connection to this, I think that, differently from what stated at lines 53-57, previous studies on agitated strings (ref 2 and ref 83 therein) had not concluded that the observed knotting probability was equal to the equilibrium one of confined semiflexible chains. Those studies had, instead, reported that a kinetic model for the stochastic braiding of nearby strand could reproduce the increase of knots with agitation time.

Answer:

Firstly, we ask reviewer 2 to read our answer to reviewer 1’s related question (3), which summarizes our understanding of the tumbling experiments. Overall, we agree with reviewer 2’s contention that the authors of [6] did not conclude “that the observed knotting probability was equal to the equilibrium one of confined semiflexible chains.” Our conclusion after re-reading the two papers is that the origin of the saturating knotting probability at long-times is unclear: it could correspond to the equilibrium knotting probability, it might also arise from kinetic (non-equilibrium) type effects such as stochastic braiding. This issue does not impact the question of whether equilibrium ideas can be used to describe our nanochannel system, because the steady-states in the two experiments are very different. In the DNA compression system the steady-state corresponds to a zero-current steady-state, which is equivalent to a state of inhomogeneous equilibrium. In the knot tumbling experiment, in contrast, finite polymer currents could exist at all times as the chain follows the revolving box, so the steady-state reached in the tumbling experiments is of a strongly non-equilibrium character.

4. The caption of Fig. 1, panel h, shows an example of a knotting event was discarded because the initially localized bright spot of the knot had become significantly more diffuse. This type of behavior has, in fact been reported in the spontaneous knotting and unknotting dynamics of flexible linear polymers. So I wonder whether the conditions need to detect knots introduce a selection bias towards tight knots. This would be a very important point to clarify for the benefit of e.g. future modeling studies seeking a quantitative agreement with the experimental measurements.

Answer:

We thank the reviewer for bringing this point to our attention. A typo in the original manuscript has raised this question; the reference to the figure in the caption should refer to the one-knot event (previously labeled g, now labeled h) and by mistake it addresses the figure below it (now labeled i, previously labeled h). Regarding knot identification, please refer to the answer to the first reviewer’s comment (number 5), where knot identification criterion is explained in detail.

In figure 1(h) in the current manuscript, during the first few seconds of relaxation two bright spots exist on the DNA molecule. However, shortly the upper one gets disentangled in the middle of the chain, indicating that the bright spot is actually an unknot. That is the reason why we do not consider it as a knot and the kymograph in figure 1(h) actually illustrates a one-knot event. We have fixed it in the caption in the manuscript.

Regarding the substance of the reviewer’s question, our knot identification criteria limits us to only counting structures that have a “spot-like” final structure as knots. Our understanding is that knots on *semiflexible* chains

should reach such a final metastable state, while the work the reviewer points us towards ([33], we think) applies to flexible chains, which do not possess the same stabilization mechanisms leading to a well-defined metastable knot size.

III. ANSWER TO REVIEWER 3

Another interesting result is the condition under which the spontaneous formation of composite knots (made mostly by two prime knots) is observed: for very high compression (i.e. for sufficiently small values of the barrier extension) this probability becomes higher than the corresponding one for a single knot. This result suggests that if the control parameters are properly tuned, it is possible to increase the complexity of the formed knots, at least within the family of composite knots. Finally it is of great interest the results on the spatial distribution of the formed knots: for a single prime knot it is shown that it preferentially forms and resides in the proximity of the barrier while, for a two-components composite knot, the distribution displays a non trivial behaviour suggesting an intriguing exclusion (topological) interaction between the two knots.

In recent years, there has been significant increase in the interest in knots in a range of different soft matter systems. This ranges from nematic liquid crystals to chemically synthesised knotted structures as well as to knotted polymers in Nature including proteins and DNA. There remain many fundamental issues about such structures including basic questions on their formation and also their dynamical behaviour. This paper should therefore be of wide interest to a number of different scientific disciplines and could attract the Nature Communications audience.

We thank the reviewer for their positive comments and we appreciate their interest in the study.

1) How the authors can be certain that the bright spots appearing during the relaxation dynamics are due to physical knots instead then, say, either to locally folded regions or to portions of the chain that are geometrically entangled but not necessarily knotted? One can imagine for instance to cases in which locally the chain seems to present a knotted portion but on larger scales it is actually unknotted. A typical example are slipknots that are known to occur very frequently in long chains and in proteins too. I think that the authors should make an effort to explain how they decide that the spots are indeed physical knots, i.e. portion of the chain that when properly closed belong to a given knot type. I think that a deeper discussion on this issue is crucial to strengthen the validity of the results presented here.

Answer:

A detailed description of the knot identification criteria and rationale has been given in response to the first reviewer (number 5). We believe that our discussion of complex unknots also applies to slip-knots. Slipknots detected on proteins [14, 34] can be untied via pulling the two ends of protein [35]. To the best of our knowledge, no slipknots have been reported for DNA molecules. In the case of slipknot formation on DNA, we believe that like folds and other entangled regions discussed, slipknots on DNA would unravel as a result of thermal fluctuations and entropic forces driving contour to less confined regions.

2) Since this study present a scheme of a knot making machine that is alternative to the one presented in Tang and co-workers (see Ref. 15), I think that a deeper comparison between the two schemes is needed to better emphasize the advantages/drawbacks of one approach with respect to the other. In Tang et al. the molecule considered is the same (T_4) but the condensation of the molecule into small regions is due to the action of an external electric field. This field induces a collapse transition of the charged DNA into a globular, isotropic, conformation. Here, instead, the hydrodynamic compression gives rise to a rather anisotropic steady state configuration, as shown by the concentration profiles reported in Figure 1 (f-i). How this difference in the starting configuration impact on the formation and complexity of knots during the relaxation? Which machine is in this respect more efficient? Which of the two protocol is more tunable and easier to control? How the knot spectrum depends on the two protocols? For example I suspect that, while in the isotropic configuration (i.e. induced by the electric field) the knots form during the relaxation are essentially prime knots (although complex), the protocol introduced here should favour the formation of composite knots, as the results seem to suggest. I think that the authors need to discuss these issues with some details, possibly in the concluding session.

Answer:

The reviewer has asked very interesting question. We can make qualitative comparisons between our approach and that proposed in [2], however there is no way to perform a rigorous comparison due to the absence of quantitative data in the literature for knotting performed via their method. Note that they report formation of single and multiple knots using the DEP compression technique, and mention that the knots form in the middle of the chains, but they do not report knotting as a function of waiting time or the knotting probability resulting from performing the experi-

ment many times with different compression conditions. Thus, we have no basis to make a comparison regarding the relative efficiency of the two approaches or the types of knots generated. In fact, one reason the kind of quantitative measurements we have performed in this study are valuable is precisely because only quantitative measurements allow the relative efficiency of differing approaches to be compared. We hope our approach might inspire similar efforts to quantify knot generation probability with other techniques so detailed comparisons such as the reviewer proposes can be made.

Regarding how the two techniques compare, we think the advantage of the nanochannel approach, from a technical point of view, is that it is in principle a parallel approach, enabling many compression and relaxation events to be performed together in parallel nanochannels. We also think that our system is physically simpler, so that in principle it should be an easier system in which to model knot-formation (with our free energy model, molecular dynamics simulation, or other analytic approaches). Our system relies purely on geometric confinement and pressure-driven low Reynolds number flow. This system gives rise to a DNA concentration profile that we can quantitatively show is consistent with a force-constrained equilibrium. The structure of the concentration profile is highly reproducible between molecules and can be simply characterized by a linear-ramp. In contrast, the approach of Tang *et al.* occurs in a strongly non-equilibrium environment, with both solvent and DNA demonstrating complex dynamics resulting from the hydrodynamic instabilities induced by the DEP force. In addition, in the approach of Tang *et al.*, the DNA is driven into a globule state, which is less well understood due to the complexity of the DEP-induced attractive interactions that drive the compression.

We have added a paragraph in the discussion section comparing the current study with that of [2]: “In a recent study, Tang *et al.* [2] introduced a technique for inducing knots on DNA molecules via application of an AC electric field. From a practical point-of-view, our approach has the advantage that it is inherently parallel; many molecules can be simultaneously compressed in an array of nanochannels and their relaxed, nanochannel-extended states analyzed. From a physical point of view, the approach of Tang *et al.* occurs in a much more complex, strongly non-equilibrium environment, with both solvent and DNA exhibiting complex dynamics resulting from the hydrodynamic instabilities induced by the DEP force. In particular, in the DEP approach the DNA tumbling dynamics leads to finite segmental current throughout the coil. In addition, the DNA is driven into a globule state, which is less well understood due to the complexity of the DEP-induced attractive interactions that drive the compression. In our system, based on geometric confinement and pressure-driven low Reynolds number flow, the DNA adopts a highly reproducible concentration profile that corresponds rigorously to an inhomogeneous equilibrium state with zero segmental current. The simplicity of the underlying DNA conformation in our system may facilitate modeling of knot generation processes. The transverse confinement in our system provides an additional parameter that can be used to tune knot-formation, with lower channel diameter in the extended de Gennes regime predicted to produce knots with greater probability at an equivalent degrees of compression (see supplementary section XI). In addition, the channel diameter likely sets an upper limit on knot size for channels below about 500 nm. Our approach may thus lead to composite knot states formed from smaller stacked prime knots distributed towards one molecule edge. In contrast, we speculate the approach of Tang *et al.* may lead to easier production of larger, topologically complex knots in the molecule center [36].”

*3) One important result on the efficiency of the knot machine is the increase of the knotting probability with the increase of the waiting time t_w . This is clearly shown in Figure 3b where t_w is reported in seconds. In order to have a feeling of how large are these time intervals, I think it would be nice to compare them with some other known time scales for DNA molecules under confinement. For example for slit-like confinement it is known that the relaxation time of the T_4 molecule within slits of widths $\sim 1/2$ microns is of the order of 2-3 seconds (see Balducci *et al.* PRL 2007). If I am not mistaken there is also a paper, in which one of the author is co-author, in which the relaxation dynamics of DNA molecules in channels has been discussed (It must be a PRL in 2005). I don't remember whether the DNA considered was a T_4 or if some characteristic time scales have been explicitly reported. Nevertheless I think that a comparison between times scale is important. Note that in this case the DNA is in a compressed state and hence its relaxation time should be larger than the one expected for a DNA that has been simply inserted in a nanochannel.*

Answer:

We agree that it is reasonable to compare the waiting times observed with other characteristic time-scales for confined DNA. We choose to compare to the extensional relaxation time of T_4 DNA in no-flow equilibrium (~ 10 s), which we estimate by scaling the relaxation times estimates for λ -DNA from [37] (the scaling can be performed by noting the relaxation time for nanochannel confined DNA scales as the square of the DNA contour length and then multiplying by the square of the contour length ratio between T_4 and λ -DNA). Regarding this point, we have added the following text to the section “Knotting probability measurement” in the manuscript: “The knotting probability rises with t_w and then asymptotes to a constant value at long-times ($t > 17$ s), suggesting a gradual equilibration of the knotting state. *This equilibration time-scale compares on order of magnitude to the extensional relaxation time of confined T_4 DNA in channels of our size in no-flow equilibrium (~ 10 s, obtained from scaling values for the λ -DNA relaxation*

time in [37] to T_4)” (italics indicates added text)

4) *How relevant is the degree of longitudinal confinement in the results presented here? Certainly the presence of the channel is crucial to establish the hydrodynamic laminar (very small Reynolds number) compressing field but is the confinement strong enough to affect also the configurational properties of the DNA in absence of the velocity field? Given the average extension in the bulk of a T4 molecule (1.3 microns) and its effective diameter it is reasonable to expect that, for the chosen channel (with average width 350 nm), the molecule in the relaxed (equilibrium) state is mildly confined i.e. at the beginning of the extended de-Gennes regime. Can the authors discuss a little bit more on this point?*

Answer: The confinement arising from the nanoscale channels used does affect the equilibrium no-flow conformation of the DNA. In particular, the no-flow extension $r_o = 14.3 \mu\text{m}$ which is greater than the $1.3 \mu\text{m}$ value for the bulk coil size the reviewer quotes. This extension is highly relevant for our study as the DNA is sufficiently linearized to enable visualization of the knots. We would agree, however, that the extension is fairly low (as we would expect for the channel size used); we estimate the fractional extension $r_o/L = 0.22$ (using $L = 63.7 \mu\text{m}$ for the dye-adjusted contour of T4). We agree that the T4 DNA is at the beginning of the extended-de Gennes regime for the channel size used.

5) *Regarding the previous point I think that the issue of the effect that confinement can have on the efficiency of the knot making machine is a very important one and maybe some additional experimental tests or at least a deeper discussion could greatly improve and strengthen the importance of the results presented so far. In this respect how difficult it would be to do some additional experiments in which either the width of the channel is reduced or, maybe simpler, a longer DNA molecule such as the lambda-DNA concatomers (the 6-lambda DNA used for example in Balducci et al. PRL 2007) is used? Of course I am not asking to redo the full analysis reported so far in the paper but maybe a rough estimate of the knotting probability for a longer DNA (or smaller channel width), would be enough to furnish some indication on how confinement can affect knot production. If, on the other hand, these additional tests would require too much time I still ask the authors to discuss this point in the text and provide an answer to this question with some arguments.*

Answer:

We feel that further experiments, while interesting, would be outside the scope of the present study. We can certainly use our model to probe the effect of varying confinement on knot-formation. We find that in the extended de Gennes regime increasing confinement is predicted to increase the probability of knot generation (see Fig. 3(a)). This is unsurprising given that increased confinement, by increasing the f_{wuk} term in the free energy will tend to enhance the free energy saved upon forming a knot, leading to a decreased single-knot free energy (Fig. 3(b), red-curve). Our model also predicts that more multiple knot states would be generated for lower channel width (Fig. 3(a)), due to the decreasing knot interaction free energy (Fig. 3(b), blue-curve), which falls for channels below 500 nm due to the decreasing knot size with lower channel dimension (the channel dimension necessarily places an upper limit on the knot size in our model). It would also be interesting to perform analogous experiments in the transition ($< 100 \text{ nm}$) and sub-persistence Odijk regimes ($< 50 \text{ nm}$). We do know that knots can be formed in channels on order of the persistence length [3], but our model will break down here as we do not expect the free energy scalings to extend to such small channels. This discussion has been added to supplementary materials, section XI.

6) *Perhaps the weakest point of the manuscript is the theoretical part where a free energy argument is used to get an estimate of the knotting probability as a function of the barrier extension (i.e. degree of compression). First of all I have found the assumption of thermal and mechanical equilibrium stated in line 240 a little bit arbitrary or at least not well explained. On which bases this assumption is made? From a rigorous point of view the compressed states are not in equilibrium. One can however assume that these are steady states and propose a sort of generalized free energy argument for a steady states.*

Answer: Our answer to reviewer 2’s third question addresses this concern in detail. In summary, our system corresponds to special type of steady state possessing zero current which is equivalent to a state of “inhomogeneous” or “force-constrained equilibrium.” Our evidence for the existence of a zero current steady-state is experimental; the profile possesses quantitative features (linear concentration ramp and scalings) that are consistent, in the context of the nonlinear convective-diffusion formalism we use to describe the compressed DNA, with a “zero current steady-state.” As the reviewer suggests, such states can be described by a generalized free energy that includes the equilibrium free energy plus work performed by external forces. This is the approach taken in the manuscript.

FIG. 3. (a) Predicted probability as a function of channel dimension of generating an event with one knot (red), two-knots (blue) and three-knot (green) on a profile with $R_b = 0.2$. Note that the large three-knot formation probability would predict that for channel widths below 300 nm, states with even more knots would be generated (although we do not include these explicitly in the calculation). The total probability of generating an event with knots is the black-curve. (b) The free energy of a single-knot state (red) and two-knot interaction free energy (blue) versus channel width. For simplicity, the channels have aspect ratio unity.

7) *Still the free energy that is considered has too many terms with free parameters that can be used to fit many possible scenarios. In this respect it is difficult to consider this theory a predictive one. Hence I suggest the authors either to add more convincing arguments on why one should believe in the predictive character of their theory or to put less emphasis on this part and saving more space for clarifying most of the unclear or poorly discussed issues mentioned above.*

Answer: Developing a quantitative model to describe knot formation is challenging due to the (current) relatively crude state of the theory. Models exist that describe nanochannel-confined knot statistics and conformation in a no-flow equilibrium situation, but in order to describe knot formation in a compressed state, e.g. a chain with $R_b < 1$, these theories must be extrapolated via scaling arguments (as discussed in section “Knot formation free energy model” of the manuscript and section V of supplementary materials). In order to compare with experiment, such a theory necessarily requires fitting, as the detailed parameters simply are not understood well enough in the compressed regime to be directly computed. In particular, the fitting parameters we use are A_b , which characterizes the bulk free energy of the knots, A_{wk} , which characterizes the knot confinement free energy and A_h which characterizes the knot friction factor. While we do not know the *exact* value of these parameters, this does *not* mean that they are allowed to take on *any* values. In particular, they must be positive and agree with unity to within an order of magnitude, which is indeed the case (our model with revised partition function gives $A_b = 1.43 \pm 0.05$, $A_{wk} = 0.98 \pm 0.12$ and $A_h = 1.12 \pm 0.07$). While we choose to fit three parameters, theory suggests fixing $A_{wk} = 1$ and $A_h = 1$ and then performing a *single parameter fit* to fix A_b , which gives agreement of equivalent quality with $A_b = 1.46 \pm 0.01$ (see Fig. 4 for the comparison between the one parameter and three parameter fits). Note that the value of A_b is most critical to the overall agreement: too low and the knotting probability is too high for all R_c , too high and the knotting probability is suppressed relative to experiment. We believe that the slightly higher value of A_b relative to unity is required because our model may underestimate the knot confinement free energy. For example, slight compression of knots at high forcing, an effect not taken into account in our model, would lead to additional

positive free energy contributions. The value of A_b might need to be increased upon fitting to compensate for the absence of these effects. We feel that these effects would be worthwhile to explore in future theoretical studies of knot-formation on compressed chains.

FIG. 4. The bold curves show the original three parameter fit yielding $A_b = 1.43 \pm 0.05$, $A_{wk} = 0.98 \pm 0.12$ and $A_h = 1.12 \pm 0.07$. The dashed curves show a one-parameter fit fixing $A_{wk} = 1$ and $A_h = 1$ and fitting A_b (the approach gives $A_b = 1.46 \pm 0.01$).

Regarding the predictive aspects of the model, we summarize the key points:

1. The model gives a quantitative mechanism explaining the increase of knotting probability with compression.
2. The model shows that interactions between knots via a hard-core repulsion mechanism preventing knot crossing can lead to suppression of multi-knot states; the degree of suppression, quantified by the interaction free energy, is correctly described by the model
3. The model captures the transition from a Poisson to a non-Poisson knot-formation regime as compression increases
4. The model predicts qualitatively correct knot spatial distributions

Overall, we feel that the model provides a quantitative framework that helps rationalize our knotting probability measurements and relate these measurements to other aspects of the experiment, such as the compressed chain concentration profiles and knot spatial distributions. This gives our hypotheses (e.g. regarding knot interactions) a more rigorous footing than if they were presented without such context. Note in particular that the model does not need to account for interactions via an additional fitting parameter. The interaction effects arise naturally from the way the model is constructed, which suggests the model is predictive (in particular, the model gives the correct interaction free energy scales, see Fig. 4b in manuscript). We feel the degree of fitting required is acceptable given the current crude state of theory and hope that our approach inspires more theoretical work in the future on problems related to knot formation in compressed chains.

We have added the following text in the second to last paragraph of section ‘‘Knot formation free energy model’’: ‘‘Equivalently, we can fix $A_{wk} = 1$ and $A_h = 1$ and perform a one-parameter fit of the parameter A_b , which yields

equivalent results (see Supplementary X)".

-
- [1] C. Micheletti and E. Orlandini, *Soft Matter* **8**, 10959 (2012).
 - [2] J. Tang, N. Du, and P. S. Doyle, *Proceedings of the National Academy of Sciences* **108**, 16153 (2011).
 - [3] R. Metzler, W. Reisner, R. Riehn, R. Austin, J. Tegenfeldt, and I. M. Sokolov, *EPL (Europhysics Letters)* **76**, 696 (2006).
 - [4] S. Bernier, A. Huang, W. Reisner, and A. Bhattacharya, submitted to *Macromolecules* (2017).
 - [5] W. Reisner, J. N. Pedersen, and R. H. Austin, *Reports on Progress in Physics* **75**, 106601 (2012).
 - [6] D. M. Raymer and D. E. Smith, *Proceedings of the National Academy of Sciences* **104**, 16432 (2007).
 - [7] D. Meluzzi, D. E. Smith, and G. Arya, *Annual review of biophysics* **39**, 349 (2010).
 - [8] X. R. Bao, H. J. Lee, and S. R. Quake, *Physical review letters* **91**, 265506 (2003).
 - [9] A. Y. Grosberg and Y. Rabin, *Physical review letters* **99**, 217801 (2007).
 - [10] L. Dai, B. Renner, C, and P. Doyle, *Macromolecules* **47**, 6135 (2014).
 - [11] L. Dai, C. B. Renner, and P. S. Doyle, *Macromolecules* **48**, 2812 (2015).
 - [12] W. R. Taylor, *Nature* **406**, 916 (2000).
 - [13] S. L. Levy, J. T. Mannion, J. Cheng, C. H. Reccius, and H. G. Craighead, *Nano Lett.* **8**, 3839 (2008).
 - [14] W. R. Taylor, *Computational biology and chemistry* **31**, 151 (2007).
 - [15] J. Cantarella and H. Johnston, *Journal of Knot Theory and its Ramifications* **7**, 1027 (1998).
 - [16] T. Biedl, E. Demaine, M. Demaine, S. Lazard, A. Lubiw, M. Overmars, S. Robbins, I. Streinu, and G. Toussaint, in *In Proc. 10th ACM-SIAM Sympos. Discrete Algorithms* (Citeseer, 1999).
 - [17] G. Toussaint, *Contributions to Algebra and Geometry* **42**, 301 (2001).
 - [18] G. A. Arteca, *Chemical Physics Letters* **328**, 45 (2000).
 - [19] Y. Arai, R. Yasuda, K.-i. Akashi, Y. Harada, H. Miyata, K. Kinoshita, and H. Itoh, *Nature* **399**, 446 (1999).
 - [20] D. Gupta, J. Sheats, A. Muralidhar, J. J. Miller, D. E. Huang, S. Mahshid, K. D. Dorfman, and W. Reisner, *The Journal of chemical physics* **140**, 214901 (2014).
 - [21] C. Zhang, F. Zhang, J. A. van Kan, and J. R. Van der Maarel, *The Journal of chemical physics* **128**, 06B611 (2008).
 - [22] F. Persson, P. Utiko, W. Reisner, N. B. Larsen, and A. Kristensen, *Nano letters* **9**, 1382 (2009).
 - [23] Y. Wang, D. R. Tree, and K. D. Dorfman, *Macromolecules* **44**, 6594 (2011).
 - [24] A. Khorshid, P. Zimny, D. Tétreault-La Roche, G. Massarelli, T. Sakaue, and W. Reisner, *Physical review letters* **113**, 268104 (2014).
 - [25] A. Khorshid, S. Amin, Z. Zhang, T. Sakaue, and W. W. Reisner, *Macromolecules* **49**, 1933 (2016).
 - [26] N. Vivek, A. R. Klotz, and P. S. Doyle, *ACS Macro Letters* **6**, 1285 (2017).
 - [27] V. V. Rybenkov, N. R. Cozzarelli, and A. V. Vologodskii, *Proceedings of the National Academy of Sciences* **90**, 5307 (1993).
 - [28] L. Bertini, A. de Sole, D. Gabrielli, G. Jona-Lasinio, and C. Landim, *Reviews of Modern Physics* **87**, 593 (2015).
 - [29] J. F. C. Kingman, *Poisson Processes* (Clarendon Press, 1992).
 - [30] L. Bertini, A. de Sole, D. Gabrielli, G. Jona-Lasinio, and C. Landim, *Journal of Statistical Physics* **135**, 857 (2009).
 - [31] J. Zhou, Y. Wang, L. D. Menard, S. Panyukov, M. Rubinstein, and M. Ramsey, *Nature Communications* **8**, 807 (2017).
 - [32] Z. Azad, M. Roushan, and R. Riehn, *Nano Letters* **15**, 5641 (2015).
 - [33] L. Tubiana, A. Rosa, F. Fragiaco, and C. Micheletti, *Macromolecules* **46**, 3669 (2013).
 - [34] T. O. Yeates, T. S. Norcross, and N. P. King, *Current opinion in chemical biology* **11**, 595 (2007).
 - [35] C. He, G. Z. Genchev, H. Lu, and H. Li, *Journal of the American Chemical Society* **134**, 10428 (2012).
 - [36] A. R. Klotz, V. Narsimhan, B. W. Soh, and P. S. Doyle, *Macromolecules* **50**, 4074 (2017).
 - [37] D. R. Tree, Y. Wang, and K. D. Dorfman, *Biomicrofluidics* **7**, 054118 (2013).

Reviewers' Comments:

Reviewer #1 Remarks to the Author:

I am satisfied with the authors responses and revisions to the paper.

Reviewer #2 Remarks to the Author:

The revised manuscript of Amin et al. contains a much improved discussion of the results. In particular, the analysis of the effective interaction between knots based on Poisson statistics is convincing and transparent.

Besides this, most other points in my previous report have been adequately dealt with and therefore I am happy to recommend publication of the manuscript.

I would still ask the authors to consider again point 4, the one regarding the possible selection bias towards tight knots, which in my view deserves an upfront discussion, though not necessarily a long one.

If I understand correctly, the author's argument is that in semiflexible chains knots must be tight. I think the results (theory + simulations) from various groups point consistently to a different conclusion. The point is best discussed by considering the probability distribution of knots of length lk , $P(lk)$.

Regardless of whether the chains are flexible or semi-flexible, the salient features of $P(lk)$ are found to be the same: (i) the modal value is largely independent of chain length, N and (ii) the decay for large lk is slow (a power law). The first feature is what one refers to as "metastability": the most probable knot length is independent on N . However, for property (ii) there is an appreciable chance that much longer knots are observed. As a matter of fact, it is known that the average (not the median) knot length increases with N for both flexible and semi-flexible chains. I think, therefore, that one should not rule out a priori the occurrence of "diffuse knots".

I perfectly understand the practical reasons for focussing on well defined "spot-like" features. However, I think it would be important to have a caveat about missing diffuse knots, especially in view of future studies.

Reviewer #3 Remarks to the Author:

After having carefully read the present manuscript, in its revised form, as well as the correspondence between editors, referees and authors, I suggest that this manuscript should be published in its present form. This is for two main reasons:

- The authors have addressed many of my previous concerns and the new version of the manuscript is greatly improved.
- Although it is still unclear to me how the theoretical results based on equilibrium free energy arguments can fully explain the out of equilibrium (although in zero current steady state) formation of knots under Poiseuille flow, the quality and the rigour of the experimental work is pretty high. This gives rise to many interesting results as well as open questions that in the near future may motivate experimentalists and theoreticians to develop new strategies to address them.

Response to Reviewers' comments:

I. ANSWER TO REVIEWER 1

I am satisfied with the authors responses and revisions to the paper.

Answer: We appreciate the reviewer's compliment and positive view of the manuscript.

II. ANSWER TO REVIEWER 2

The revised manuscript of Amin et al. contains a much improved discussion of the results. In particular, the analysis of the effective interaction between knots based on Poisson statistics is convincing and transparent. Besides this, most other points in my previous report have been adequately dealt with and therefore I am happy to recommend publication of the manuscript.

Answer: We thank the reviewer for their positive comments. The reviewer's constructive comment on Poissonian statistics in the previous round of review led to a greatly improved paper, which we appreciate.

I would still ask the authors to consider again point 4, the one regarding the possible selection bias towards tight knots, which in my view deserves an upfront discussion, though not necessarily a long one. If I understand correctly, the author's argument is that in semi-flexible chains knots must be tight. I think the results (theory + simulations) from various groups point consistently to a different conclusion. The point is best discussed by considering the probability distribution of knots of length lk , $P(lk)$. Regardless of whether the chains are flexible or semi-flexible, the salient features of $P(lk)$ are found to be the same: (i) the modal value is largely independent of chain length, N and (ii) the decay for large lk is slow (a power law). The first feature is what one refers to as "metastability": the most probable knot length is independent on N . However, for property (ii) there is an appreciable chance that much longer knots are observed. As a matter of fact, it is known that the average (not the median) knot length increases with N for both flexible and semi-flexible chains. I think, therefore, that one should not rule out a priori the occurrence of "diffuse knots". I perfectly understand the practical reasons for focussing on well defined "spot-like" features. However, I think it would

be important to have a caveat about missing diffuse knots, especially in view of future studies.

Answer:

We agree that there is no consensus of opinion on the tightness of knots among different studies. While many experimental [1–5] and theoretical [6–12] studies agree that knots on chains are localized as tight knots, other studies [13–15] believe that knots can form which are not tightly localized and can spontaneously expand along chains. From an experimental point of view, we do not observe stable structures along a chain in no-flow equilibrium that might be considered “diffuse” knots, that is bright structures that occupy a large fraction of the molecule extension and appear to undergo a high degree of thermal fluctuation. We think the large confinement free energy cost of having a higher than equilibrium DNA concentration would tend to enhance the tightening effect, driving DNA from the diffuse knot to other regions of the chain. An important caveat here is that this above observation applies only to no-flow equilibrium; we can say nothing about the situation regarding diffuse knots during chain relaxation from the compressed state. In particular, we cannot rule out that diffuse knot configurations, particularly if they are close to the chain edges, might unravel during chain relaxation and be missed. Thus, we have added a few sentences to the discussion section in the manuscript:

“While many experimental [1–5] and theoretical [6–12] studies agree that knots on chains are localized as tight knots, others [13–15] believe that knots can form which are not tightly localized and can spontaneously expand along chains. In our knot detection criteria, knots are persistent, localized and bright features representing metastable tight knots. While we do not observe diffuse knot configurations at no-flow equilibrium for long waiting times after chain relaxation, there might be some diffuse knotted configurations that we have missed in our knot numeration because they might unravel at short times during the relaxation process, especially if they are close to the molecule edges. Future Brownian dynamics simulations might elucidate the evolution of knots during molecule relaxation from the compressed state and estimate how many knots might be lost during this process.”

III. ANSWER TO REVIEWER 3

After having carefully read the present manuscript, in its revised form, as well as the correspondence between editors, referees and authors, I suggest that this manuscript should be published in its present form. This is for two main reasons: The authors have addressed many of my previous concerns and the new version of the manuscript is greatly improved.

Answer: We are grateful to the reviewer for their supportive view on the manuscript.

Although it is still unclear to me how the theoretical results based on equilibrium free energy arguments can fully explain the out of equilibrium (although in zero current steady state) formation of knots under Poiseuille flow, the quality and the rigour of the experimental work is pretty high. This gives rise to many interesting results as well as open questions that in the near future may motivate experimentalists and theoreticians to develop new strategies to address them.

Answer: We agree with the reviewer that our equilibrium theory does not “fully” explain knot-formation. The problem here is that in order to have a “full” explanation of why an equilibrium theory can be applied, one must understand why the relevant free energy barriers for knot-formation are low. While we know experimentally that knot-formation kinetics are favourable (e.g. see Fig. 3b), *we do not understand the physical mechanism behind the lowered barriers we observe.* In our view, understanding this mechanism should be a key direction for future work, one of the “open questions” this paper raises and that the reviewer alludes to as being part of the work’s impact. In the discussion we state: “A complete understanding of knot formation in our system requires understanding the physics behind the lowered topological barriers leading to favorable kinetics at experimentally accessible time-scales.” To emphasize that we do not at all have a complete understanding, in agreement with the reviewer, we rewrite this sentence with a conditional phrasing: “A complete understanding of knot formation in our system *would require* understanding the physics behind the lowered topological barriers leading to favorable kinetics at experimentally accessible time-scales.” The two paragraphs that follow this statement go into detailed speculation for why the

barriers may be lower in our system upon chain compression.

- [1] E. Ercolini, F. Valle, J. Adamcik, G. Witz, R. Metzler, P. De Los Rios, J. Roca, and G. Dietler, *Physical review letters* **98**, 058102 (2007).
- [2] X. R. Bao, H. J. Lee, and S. R. Quake, *Physical review letters* **91**, 265506 (2003).
- [3] R. Metzler, W. Reisner, R. Riehn, R. Austin, J. Tegenfeldt, and I. M. Sokolov, *EPL (Europhysics Letters)* **76**, 696 (2006).
- [4] J. Tang, N. Du, and P. S. Doyle, *Proceedings of the National Academy of Sciences* **108**, 16153 (2011).
- [5] A. R. Klotz, V. Narsimhan, B. W. Soh, and P. S. Doyle, *Macromolecules* **50**, 4074 (2017).
- [6] V. Katritch, W. K. Olson, A. Vologodskii, J. Dubochet, and A. Stasiak, *Physical Review E* **61**, 5545 (2000).
- [7] A. Y. Grosberg and Y. Rabin, *Physical review letters* **99**, 217801 (2007).
- [8] L. Dai, C. B. Renner, and P. S. Doyle, *Macromolecules* **47**, 6135 (2014).
- [9] L. Dai, C. B. Renner, and P. S. Doyle, *Physical review letters* **114**, 037801 (2015).
- [10] L. Dai, C. B. Renner, and P. S. Doyle, *Macromolecules* **48**, 2812 (2015).
- [11] P. G. Dommersnes, Y. Kantor, and M. Kardar, *Physical Review E* **66**, 031802 (2002).
- [12] E. Guitter and E. Orlandini, *Journal of Physics A: Mathematical and General* **32**, 1359 (1999).
- [13] W. Mobius, E. Frey, and U. Gerland, *Nano letters* **8**, 4518 (2008).
- [14] L. Tubiana, A. Rosa, F. Fragiaco, and C. Micheletti, *Macromolecules* **46**, 3669 (2013).
- [15] X. Zheng and A. Vologodskii, *Physical Review E* **81**, 041806 (2010).